# How can we understand the global distribution of the solar cycle signal on the Earth's surface?

Kunihiko Kodera[1], Rémi Thiéblemont[2], Seiji Yukimoto[3], Katja Matthes[4,5]

[1]Institute for Space-Earth Environmental Research, Nagoya University, Nagoya, 464-8601 Japan
[2]Laboratoire Atmosphères Milieux Observations Spatiales, Guyancourt, 78280, France
[3]Meteorological Research Institute, Tsukuba, 305-0052  Japan
[4]Research Division Ocean Circulation and Climate, GEOMAR Helmholtz Centre for Ocean Research, Kiel, 24105 Germany
[5]Christian-Albrechts Universität zu Kiel, Kiel, 24105 Germany

*Correspondence to*: Kunihiko Kodera (kodera@isee.nagoya-u.ac.jp)

**Abstract.** To understand solar cycle signals on the Earth's surface and identify the physical mechanisms responsible, surface temperature variations from observations as well as climate model data are analyzed to characterize their spatial structure. The solar signal in the annual mean surface temperature is characterized by i) mid-latitude warming and ii) no overall tropical warming. The mid-latitude warming during solar maxima in both hemispheres is associated with a downward penetration of zonal mean zonal wind anomalies from the upper stratosphere during late winter. During Northern Hemisphere winter this is manifested in a modulation of the polar-night jet whereas in the Southern Hemisphere the upper stratospheric subtropical jet plays the major role. Warming signals are particularly apparent over the Eurasian continent and ocean frontal zones, including a previously reported lagged response over the North Atlantic. In the tropics, local warming occurs over the Indian and central Pacific oceans during high solar activity. However, this warming is counter balanced by cooling over the cold tongue sectors in the southeastern Pacific and the South Atlantic, and results in a very weak zonally averaged tropical mean signal. The cooling in the ocean basins is associated with stronger cross-equatorial winds resulting from a northward shift of the ascending branch of the Hadley circulation during solar maxima. To understand the complex processes involved in the solar signal transfer, results of an idealized middle atmosphere–ocean coupled model experiment on the impact of stratospheric zonal wind changes are compared with solar signals in observations. Model integration of 100 years of strong or weak stratospheric westerly jet condition in winter may exaggerate long-term ocean feedback. However, the role of ocean in the solar influence on the Earth's surface can be better seen. Although the momentum forcing differs from that of solar radiative forcing, the model results suggest that stratospheric changes can influence the troposphere not only in the extra-tropics but also in the tropics through i) a downward migration of wave–zonal mean flow interactions and ii) changes in the stratospheric mean meridional circulation. These experiments support earlier evidence of an indirect solar influence from the stratosphere.

# 1 Introduction

The influence of solar activity on the Earth's surface, especially that of the 11-year solar cycle, has been debated for a long time (e.g., Pittock, 1978; Legras, 2010). The climate impact of solar influence is generally assessed in terms of the radiative forcing (e.g., IPCC, 2013). Recent direct measurements from space reveal that changes in the total solar irradiance (TSI) associated with the 11-year solar cycle are about 0.1% (1.3 W m$^{-2}$) (Kopp and Lean, 2011). Such small variations are not expected to have a significant impact on surface climate, and so several mechanisms have been proposed that act to amplify the initially small solar effects. One amplification mechanism is enhancement of the direct TSI effect at the ocean surface due to a feedback of water vapor transport in the tropical Pacific (Meehl et al., 2008, 2009). Another possible amplification mechanism works through a change in the solar spectrum, in particular in the ultra-violet (UV) range, directly affecting the stratopause region and enhancing temperatures and ozone concentrations during solar cycle. The amplification and the downward penetration of the small initial solar signal occur through stratospheric dynamical processes (e.g., Kodera and Kuroda, 2002). The impact of cosmic rays on surface temperature through changes in cloud cover has also been proposed (Svensmark and Friis-Christensen, 1997).

Besides apparently small direct solar effect, another problem of explaining solar influence on climate is the rather unstable relationship between the 11-year solar cycle and the Earth's global mean surface temperature, as a breakdown or even the reversal of the relationship occurs during different time periods (e.g., Nitta and Yoshimura, 1993; Georgieva et al., 2007; Souza-Echer, 2012). However, Zhou and Tung (2010) extracted a global spatial pattern of sea surface temperature (SST) variations associated with the solar cycle by applying a composite mean difference (CMD) projection method, particularly relevant to estimate the robustness of a global spatial signal. This method segregates data into groups of high and low solar activity during the 11-year cycle. A global spatial pattern is then obtained from the composite-mean difference between the high and low solar group. Finally, the original data is projected onto this CMD spatial pattern, resulting in a time series. The method is successful when the correlation between the resulting time series and the solar forcing is high. They demonstrated that the coefficients of this CMD pattern projected onto the global SST field show a steady and highly robust relationship with the solar activity over more than 10 solar cycles (represented by the TSI for the past 153 years reconstructed by Wang et al., 2005). This indicates that a global spatial pattern, rather than a globally averaged temperature, is crucial to understanding solar influences at the surface.

Various studies of the solar influence on weather and climate were reviewed by Gray et al. (2010). Here, we do not attempt to extensively review previous works, but rather find consistent aspects of the solar signals reported in many independent studies. The surface response to solar forcing is regionally distributed; that means that the solar signal is influenced by the internal dynamics of the climate system. Our paper aims at suggesting supplementary physical processes that may help to better

understand the global distribution of the solar signal. In the following, we will particularly show that the atmosphere-ocean interactions may play an important role to characterize surface solar signal over baroclinic zones and tropical cold tong regions.

Surface temperature and pressure have been measured for more than 100 years. Thus, the relationship between surface temperature variations and solar activity can be investigated using a global historical dataset. Because sea surface temperature (SST) is more persistent than the sea-level pressure (SLP), long-term variations can be more easily detected in the temperature field. Therefore, we investigate mainly surface temperature variation from the historical data, complemented by pressure or geopotential height fields with a modern dataset. Direct measurement of the solar UV is only recent, but a record of the sunspot number, which is a proxy of the solar extreme ultraviolet (EUV), is available from the 18[th] century. The solar EUV produce the ionization in the Earth's upper atmosphere. Therefore, change in the solar EUV radiation is felt on the Earth's surface as change in geomagnetic field induced by the electric current in the ionosphere. It is, thus, possible to associate the variation of sunspot number to the solar EUV activity. Comparison of the variation calculated from Earth's magnetic field demonstrates excellent agreement between the 10.7 cm solar radio flux (F10.7) and the sunspot number (Svalgaard, 2007). Therefore, both can be used as a proxy of the solar irradiance variation.

Annual mean surface temperature anomalies related to the solar cycle have been studied using various methods and different historical global datasets covering between 120 and 150 years. Lohmann et al. (2004) calculated the correlation coefficient between the proxy solar irradiance from Lean et al. (1995) and band-pass (9−5 year period) filtered SSTs reconstructed by Kaplan et al. (1998) from 1856 to 2000. Lean and Rind (2008) extracted solar signals by applying a multiple linear regression analysis to surface temperatures reconstructed by the University of East Anglia Climatic Research Unit F (Brohan et al., 2006) for the period 1889–2006. A similar multiple linear regression analysis was conducted by Tung and Zhou (2010), who compared the regression analysis of two different historical datasets, namely NOAA's Extended Reconstructed Sea Surface Temperatures (ERSST) and the Hadley Centre Sea Ice and Sea Surface Temperature (HadISST) dataset (Rayner et al., 2003), to confirm consistent features of the solar signal. Gray et al. (2013) performed a lagged multiple linear regression analysis to investigate delayed components in the solar signal using the HadISST dataset. Despite different reconstructions and analysis methods, common features are seen during high solar activity in the surface temperatures: a mid-latitude warming, and a tropical cooling in the southeastern Pacific and the South Atlantic. Note that this cooling is different from the La Niña-like pattern previously reported (van Loon et al., 2007; Meehl et al., 2008, 2009) and will be discussed in more detail below.

We first compare the analysis results of a historical surface temperature dataset with those of a modern dataset to identify the fundamental global features of surface temperature variations related to the solar cycle; i.e., the observed surface solar signals. Next, we study the vertical structure of the solar signal with recent data to identify the physical mechanisms producing the surface solar signals. Identification of the causes and characteristics of solar signals is particularly difficult for decadal-scale periodic variations because strong feedbacks exist on these timescales in the climate system. To better understand the

mechanisms producing the surface solar signal, we revisit results from an idealized middle atmosphere–ocean coupled general circulation experiment where a momentum forcing has been applied in the stratosphere (Yukimoto and Kodera, 2007).

The remainder of this paper is organized as follows. After describing the data and method of analysis in section 2, characteristics of the solar signal in atmospheric as well as oceanic variables are described in section 3. To understand the complex processes for the solar signal transfer involving stratosphere–troposphere–ocean coupling, results of an idealized numerical experiment are compared with observed solar signals in section 4. To get insight into a centennial solar variation such as the Maunder minimum, the effect of centennial scale stratospheric circulation changes on the troposphere is briefly studied in section 5. Finally, discussions and summary about the possible mechanisms producing the solar influence on the Earth's surface are given in section 5.

## 2 Data and Analysis

### 2.1 Data

This study combines the analysis of a historical SST dataset to characterize the surface response to the 11-year solar cycle, with a modern reanalysis dataset to investigate the underlying dynamical processes. For the historical dataset, we use the NOAA Extended Reconstructed SST v3b (ERSST), described by Smith et al. (2008) and available at http://www.esrl.noaa.gov/psd/data/gridded/data.noaa.ersst.html. The ERSST dataset spans more than 160 years from 1854 to the present, with monthly resolution, and a spatial resolution of 2° longitude × 2° latitude from 88°N to 88°S and 0°E to 358°E. Note that, however, data are sparse before 1880. To examine the tropospheric and stratospheric dynamical response to the solar cycle, we use the ERA-Interim atmospheric reanalysis produced by the European Centre for Medium-Range Weather Forecasts (ECMWF) (Dee et al., 2011). We used the ERA-Interim (ERA-I) dataset from 1 January 1979 to 2010. In this study, we used monthly mean data, provided on 23 pressure levels from 1000 hPa to 1 hPa with a spatial resolution of 2.5° longitude × 2.5° latitude.

### 2.2 Multiple linear regression model

Following numerous earlier studies (e.g., Lean and Rind, 2008; Frame and Gray, 2010; Chiodo et al., 2014; Mitchell et al., 2015a,b), the ocean and atmosphere responses to solar variations are examined using a multiple linear regression model (MLR). This technique can isolate the effects of different forcings, represented by explanatory variables (or regressors), on the variance of a time-dependent variable (or predictand). Annual signals are extracted by applying the MLR to continuous monthly resolved time series. Monthly or seasonal signals (two to three consecutive months) are diagnosed by applying the MLR to time series of the individual month or season (i.e., the seasonal average is performed prior to the MLR), respectively. All data time series have the seasonal cycle removed before the MLR, as well as before any seasonal-average calculations.

The MLR model is applied at each location and is given by

$$X(t) = A \cdot CO_2(t) + B \cdot N3.4(t) + C \cdot F10.7(t - \Delta t) + D \cdot AOD(t) + E \cdot QBOa(t) + F \cdot QBOb(t) + \epsilon(t), \qquad (1)$$

where $X$(t) is the time dependent variable, the first six terms on the right-hand side of the equation correspond to the product of one time-dependent explanatory variable (e.g., $CO_2(t)$) and its regression coefficient (e.g., A), and the last term ε($t$) is the residual error.

The explanatory variables considered for the MLR describe variability sources that are demonstrated to have a significant impact on the surface, troposphere and middle atmosphere dynamics and have been broadly used in solar-climate studies based on model and reanalysis (e.g. Chiodo et al., 2014; Mitchell et al., 2015a,b). The explanatory variables are defined as follows: the CO₂ concentration (Meinshausen et al., 2011) (available at http://climate.uvic.ca/EMICAR5/forcing_data/RCP85_MIDYR_CONC.DAT) to account for the increase in anthropogenic forcing; the Nino 3.4 index derived from the ERSST v3b dataset; the F10.7 cm solar radio flux index (available at http://lasp.colorado.edu/lisird/tss/noaa_radio_flux.html); and the global aerosol optical depth (AOD) at 550 nm updated from Sato et al. (1993) to represent volcanic effects and two stratospheric quasi-biennial oscillation (QBO) orthogonal indices (QBOa and QBOb) defined as the first two principal components of the ERA-I zonal mean zonal wind in the latitude interval (10°S, 10°N) and pressure–height interval (70− 5) hPa, respectively.

QBO regressors and F10.7 index are available only from mid-20th centuries, so that QBO regressors are not included in the MLR and the F10.7 index is replaced by the sunspot numbers when long-term historical SST dataset is analyzed. Sensitivity tests of the MLR model revealed that including or removing the stratospheric QBO regressors for the period 1979-2010 negligibly affects the solar regression coefficients and their statistical significance, in particular in the troposphere. Although the F10.7 cm index more directly represents the irradiance variability in the UV band than the sunspot number (Tapping, 2013), both indices concur at annual timescales: a correlation coefficient of 0.997 between the annually averaged F10.7 and sunspot number time series is found for the period 1965-2012. The solar regression coefficient used in our study assumes that a difference of 130 solar flux units (1 sfu = $10^{-22}$ W m$^{-2}$ Hz$^{-1}$) or 100 sunspots represents the difference between the 11-year solar cycle maximum and minimum.

To investigate the effect of the ocean memory on the surface response to solar variability (e.g., Gray et al., 2013; Thiéblemont et al., 2015), we calculated the MLR at different time lags (Δt in months or years) with respect to the solar regressor. The Arctic Oscillation (AO) or the North Atlantic Oscillation (NAO) is climate mode which is partly driven by solar variability as will be shown later. Hence, it is not appropriate to include its index in a MLR model which aims at examining the solar cycle effect on surface climate.

When applying regression techniques, it is essential to carefully consider possible autocorrelation in the residual to assess statistical significances of the regression coefficients. Autocorrelation in the residual leads to an underestimation of the regression coefficient uncertainties, and thus a narrowing of the confidence intervals. A common method employed to circumvent the residual autocorrelation problem is to treat the residual term as an autoregressive process (Tiao et al., 1990). The first step of the procedure, also called prewhitening, consists of correcting both the predictors and the predictand ($X$) with the autocorrelation coefficient of the residual term estimated from a first application of the regression model. The prewhitening procedure is then repeated on the modified predictors and predictand until the residual is no longer significantly autocorrelated. The statistical significance of the autocorrelation is assessed with a Durbin−Watson test. The application of the Tiao et al. method can be found in several papers examining the solar signal (e.g., Austin et al., 2008; Mitchell et al., 2015b). We generally found that a single application of the prewhitening procedure was sufficient to remove the residual autocorrelation almost completely (more than 95% of the grid points). Once the prewhitening step has been performed, the statistical significance of the regression coefficients is calculated using a two-tailed Student's $t$-test.

The use of the MLR approach to separate the contribution of different factors on climate variability has inherent limitations that should be kept in mind when analyzing the results. The MLR particularly relies on several assumptions that may not be valid in all cases. However, composite analysis of the monthly mean data based on high and low levels of solar activity (e.g., Kuroda and Kodera, 2002; Lu et al., 2011) also produces similar solar signals as those obtained from the MLR method (Figs. 6, 7). Therefore, in spite of the limitations, the MLR method may be useful to get approximate solar signals (Kuchar et al., 2015). We note that highly non-linear responses can be produced through the interaction between different forcings: for example, between El Niño–Southern Oscillation (ENSO) and solar signals (Marsh and Garcia, 2007), solar and QBO signals (Matthes et al., 2013), as well as volcanic and solar signals (Chiodo et al., 2014). Usually this kind of interaction occurs at a specific location and time, which needs to be investigated in separate studies. Sophisticated attribution methods which can account for non-linearity have been used, such as machine learning methods (Blume et al, 2012), or optimal detection (Stott et al, 2003; Mitchell, 2016). Although these methods allow advanced statistics to be established, their limited interpretive capacities make it difficult to study physical mechanisms.

## 3 Solar signal

### 3.1 Surface temperature signal

As mentioned in the introduction, Zhou and Tung (2010; hereafter ZT2010) calculated CMDs between high- and low-activity periods of the 11-year solar cycle using the ERSST dataset. In their analysis, data near the World War II period (1942−1950) were excluded. We performed the same CMD analysis using the same dataset as ZT2010, but included all data from 1854 to 2007. We confirm the results of ZT2010 in Fig. 1a. The correlation coefficient between the expansion coefficients of the extracted pattern and the solar index shows a similar high correlation (0.69).

To assess the stability of the relationship between SSTs and the solar cycle, the use of a long dataset is crucial. However, historical datasets have problems with spatial coverage and inhomogeneity of the observing systems. This drawback may be compensated by a comparison with a recent global dataset assimilating satellite observations. Figure 1b shows the surface solar signal extracted by MLR using the ERA-I and F10.7 cm radio flux time series (solar index) as one of the explanatory variables for the period from 1979 to 2010. Despite the short time period of only 3 solar cycles, the results show a similar pattern in surface temperature to those obtained from longer historical datasets. Common features in the spatial structure of the solar signal in surface temperatures include i) (sub-polar regions): warming around 45°−60°N over the Eurasian continent and cooling over west of Greenland; ii) (mid-latitudes): warming over the ocean basins around 30°−45° latitudes in the Northern Hemisphere (NH) as well as in the Southern Hemisphere (SH); iii) (tropics): warming over the Indian Ocean and the central Pacific, and cooling in the East Pacific and the Atlantic, particularly in the SH. These characteristics are also found in a number of other studies cited in the introduction that use different analysis techniques. In spite of overall similarities, large differences can be found in some regions, such as over the subtropical eastern Pacific east of Hawaii. It shows large warming in the historical data (Fig. 1a), but cooling in the modern era data (Fig. 1b). It is however difficult to identify whether the difference in short-term data is merely due to statistical fluctuations, or related to a change in basic climatological states caused by other factors, such as global warming, or ocean circulation change, etc. Here, we concentrate on the stable solar response to first understand how it is produced at the Earth's surface. To investigate the solar signals over the ocean basins specifically, equatorward gradients of climatological SSTs are shown in Fig. 1c. The regions where warming during solar maxima occurs roughly correspond to regions of strong meridional SST temperature gradients. The case of solar signals over the North Atlantic frontal zone is more complicated (see Fig. 1a), and in fact solar signals over the North Atlantic are delayed by about 3-years (Gray et al., 2013; Scaife et al., 2013; Andrews et al. 2015; Thiéblemont et al., 2015), and will be discussed later. Note also that regions with cool solar signals in the tropics coincide with sectors of the cold tongue over the equatorial East Pacific and the Atlantic. This kind of temperature pattern is quite different from the expected impact of TSI variations from an energy balance model. Stevens and North (1996) estimated a warming in the tropics from such a model, in particular over the continents.

To identify the physical mechanisms responsible for the surface solar signals, a comparison of the surface temperature pattern associated with other forcings has been performed similarly to Lean and Rind (2008). The zonal-mean surface temperature pattern extracted by a MLR is shown in Fig. 2. In the MLR model of Lean and Rind (2008), who used historical data, anthropogenic forcing combines anthropogenic aerosol and greenhouse gases effects. Although their anthropogenic signal shows a rather uniform warming latitudinally which differs from that of the present study, the other signals (i.e. ENSO, volcanic and solar) are similar. ENSO related temperature variations (Fig. 2a) are confined to the tropics. The response to volcanic aerosol (Fig. 2b) is a global cooling, whereas the response to anthropogenic greenhouse gas forcing (Fig. 2c) is characterized by a large warming in the polar region of the NH. A cooling trend is also found in the Southern Ocean around 60°S. However,

it could also result from ozone depletion (Thompson et al., 2011), because trends in $CO_2$ and ozone concentration cannot not be well separated due to the short analysis period. The solar signal is large in mid-latitudes in both hemisphere (Fig. 2d) as already reported in Lean and Rind (2008). The reason for such a latitudinal distribution has not been addressed, however. If the surface solar signal originates from the absorption of the solar energy at the Earth's surface, we should expect higher solar

signal in the tropics similar to the climatological SST distribution (Fig. 2e). However, large solar signal is observed in the frontal zones where the meridional gradient of surface temperature is the largest (Fig. 2f) and where the interaction between the atmosphere and the ocean is particularly strong (Nakamura et al., 2008). This suggests thus a possible role of atmosphere–ocean interaction in the baroclinic zone in the modulation and amplification of the solar signal.

To explain solar signals at the surface in high latitudes, the role of the Annular Mode (AM) (Thompson and Wallace, 2000) in the NH (NAM) in mediating tropospheric solar signals has been suggested (e.g., Baldwin and Dunkerton, 2005). The AM in the Southern Hemisphere (SH) is called as the SAM. However, in the SH Lu et al. (2011) found little relationship between the solar cycle and the SAM on the surface. The surface signal of NAM and SAM are also called Arctic Oscillation (AO) and Antarctic Oscillation (AAO), respectively. The question we address here is the role of AM in a global perspective: How does

the solar signal manifest in the SH and in the NH? Figure 3 compares solar signals with annular modes in the two hemispheres. In NH winter (DJF), solar signals exhibit a similar pattern to the NAM: a warming over the Eurasian continent and the ocean basins along 30°N–45°N latitudes, and a cooling west of Greenland. Stronger westerly winds associated with the NAM and surface solar signals occur at lower latitudes over the American continent than over the Eurasian continent. This means that the NAM is not strictly annular, but also contains a stationary planetary wave structure. It should be noted that the spatial

pattern of the solar signal is similar to that of the NAM. In SH spring (SON), solar signals are characterized by a warming in mid-latitudes associated with anomalous westerlies around 40°S –50°S. However, the SAM pattern typically involves a strong warming around the Antarctic Peninsula and the southern tip of the South American continent (Thompson and Wallace, 2000; Gillett et al., 2006) in association with anomalous westerlies near the polar region around 55°–65°S. This is the reason why solar signals are not projected to the SAM in the SH, unlike to the NH.

## 3.2 Zonal-mean vertical structure

Since the surface solar signals during the recent (1979–2010) period is very similar to that of the longer historical period (1854–2007) (Fig. 1), we may gain further insight into the processes responsible for the solar signal transfer from the stratosphere to the troposphere and the ocean by analyzing the modern dataset in more detail. First of all, it should be noted

that there are two kinds of westerly jets in the middle atmosphere. Climatological poleward temperature gradient during winter solstice (Jun in the SH, and December in the NH) is displayed in Fig. 4. The meridional temperature gradient is large in the subtropics of the upper stratosphere due to solar UV heating, while in the lower stratosphere, the gradient is strong around the

polar night region due to longwave cooling. They are respectively connected to the subtropical and polar night jet. From this, we can expect that the variation in the solar UV heating first manifests in the subtropics of the stratopause region.

Figure 5 shows solar signals in the annual-mean a) zonal mean zonal wind, b) zonal mean air temperature, and c) pressure coordinate vertical velocity in the tropical troposphere using the same MLR analysis as in Fig. 1b. The results of similar MLR analyses using meteorological reanalysis data have also been published (e.g., Frame and Gray, 2010; Chiodo et al., 2014; Mitchell et al., 2015a). During periods of high solar activity, warming signals appear at three levels: the upper stratosphere–stratopause (5−1 hPa), the lower to middle stratosphere (100−20 hPa), and the troposphere (1000−300 hPa) (Fig. 5a). The warming around the stratopause extends globally from the tropics to the polar regions, while the warming in the lower stratosphere is confined to the tropics. The associated stronger meridional temperature gradient in the subtropical upper stratosphere is connected, by the thermal–wind relationship, to enhanced subtropical jets around 30°−40° latitude in both hemispheres in the upper stratosphere (Fig. 5b). Stronger subtropical jets extend farther to lower altitudes in association with a warming in the tropical lower stratosphere. The narrow latitudinal extent of the zonal mean zonal wind anomalies at mid-latitudes of the middle-lower stratosphere in Fig. 5 is difficult to explain only from a radiative heating change. The differences in the latitudinal structure of the warming suggest that the warming in the stratopause–upper stratosphere has a radiative origin, while for the second warming in the middle to lower tropical stratosphere, dynamical process plays an important role as suggested in previous studies (e.g. Kodera and Kuroda, 2002; Hood and Soukharev, 2012).

In the troposphere, a statistically significant warming occurs in the extra-tropics around 40°−45° latitude in both hemispheres (Fig. 5a), similar to that of the surface temperature anomalies in Fig. 1. Warming also occurs over Antarctica in association with a weakening of the high-latitude westerly flow. Note that there is practically no warming in the entire tropical troposphere from the surface to the tropopause. This does not mean that there is no solar influence in this region, but temperature variations in the tropical troposphere are generally small due to feedback with convective activity (Eguchi et al., 2015). Therefore, the response in vertical velocity is crucial in the tropical troposphere, although it is not directly measured. Solar signals in the vertical velocity are generally downward around the equator, but upward in off-equatorial regions around 15°−20° latitude (Fig. 5c). Note also that solar signals in the zonal mean zonal wind are symmetric around the equator in the upper stratosphere (Fig. 5b). However, in polar regions the zonal mean winds in the lower stratosphere differ markedly between the NH and SH. This can be seen more clearly as differences in the seasonal march in Fig. 6 for monthly solar signals in zonal mean winds during SH and NH winter. In early winter, the subtropical jet develops in the upper stratosphere in both hemispheres. In the NH, anomalous westerlies shift poleward and downward to the troposphere, and the stratospheric polar-night jet weakens significantly in February. In the SH, however, the poleward shift is small and the strong anomalous westerlies descend in the mid-latitude troposphere, forming a pair of westerly and easterly zonal mean wind anomalies at high latitudes in September.

Solar signals in zonal mean temperature and extracted by the MLR are shown in Fig. 7 (zonal mean zonal winds are also plotted in Fig. 7a with green lines). The lower stratospheric tropical warming occurs during a period when the stratospheric subtropical westerly winds develop, in July–August in the SH and in November–December in the NH. A tropospheric warming in mid-latitudes occurs in September−October in the SH and in January−February in the NH, and is associated with the

downward penetration of westerly zonal mean wind anomalies from the stratosphere (Fig. 6). The differences in the latitudinal structure of surface solar signals in Fig. 3 are consistent with the differences in the downward penetration in the two hemispheres: downward penetration occurs through a modulation of the polar-night jet in the NH that projects onto the NAM, but a modulation of the subtropical jet in the SH does not project onto the SAM.

### 3.3 Interactions with the ocean

The role of the ocean as heat capacitor can be seen as persistent surface temperature anomalies from winter to spring. In addition, ocean currents advect SST anomalies to higher latitudes, which may introduce further delayed response. The evolution from winter to spring of the solar signals in surface temperatures in the mid-latitudes of the NH is illustrated in Fig. 8a and 8b, respectively. In winter, stratospheric zonal mean zonal wind anomalies extend from the stratosphere to the troposphere, and lead to a seesaw pattern between the polar region and mid-latitudes, similar to the NAM as shown in Fig. 3.

In spring, stratospheric circulation anomalies associated with the polar nigh jet start vanishing. This coincides with a weakening of the temperature anomalies over the continents. Conversely, temperature anomalies over the ocean basins east of the continents not only persist from winter but also continue to develop. The positive temperature anomalies over the North Pacific east of Japan extend along 40°N. In the Atlantic sector, positive temperature anomalies are located at lower latitudes along the southeastern US coastal region. A similar SST response in spring has been confirmed with a longer historical SST dataset from

1882 to 2008 (see Figure 4 of Tung and Zhou, 2010). Note that temperature anomalies in the Pacific sector are found around ocean frontal zones, where meridional temperature gradients are strong (Fig. 8c), but in the Atlantic sector temperature anomalies are located at lower latitudes.

As noted by Zhou and Tang (2010), mid-latitudes Northwestern Atlantic is cold during periods of high solar activity. However,

positive SST anomalies located in the subtropics of the Atlantic sector in the east America shift gradually northward with time. Lagged solar signals in the SSTs are show in Fig. 9 for (a) modern and (b) historical periods. In both cases, when SST anomalies reach the sub-Arctic frontal zone (around 45°N) with a 2-3 years lag, a meridional dipole pattern similar to the NAO develops (Gray et al., 2013; Scaife et al., 2013; Andrews et al., 2015; Thiéblemont, 2015). Although the cooling at mid-latitudes appears less pronounced in the modern era, the northward shift of positive anomalies associated with the solar cycle over North Atlantic

remains unchanged.

### 3.4. Tropical solar signals

Figure 10 shows the tropical part of the solar signal in SSTs extracted from the global picture in Fig. 1a. As mentioned previously, this pattern is characterized by a cooling over the East Pacific and the Atlantic in the SH and a warming in the central Pacific. To identify the characteristics of the spatial structure of these variations, an empirical orthogonal function (EOF) analysis is conducted on the SSTs over the tropical Pacific and the Atlantic sectors during September through February when ENSO shows the greatest persistence (Wolter and Timlin, 2011). For the EOF analysis, the period 1890-2012 is chosen for better observational coverage (Smith and Reynolds, 2003). The leading and the second EOFs represent canonical ENSO and secular trends, respectively. The third EOF shows decadal variations and its spatial pattern is illustrated in Fig. 10b. The solar signal (Fig. 10a) agrees well with the spatial structure of EOF3, which is characterized by a cooling over the cold tongue regions and a warming over the warm pool region in the central Pacific. This pattern of tropical SSTs, known as El Nino Modoki, has been extracted as EOF2 with a shorter dataset from 1979 through 2004 (Fig. 2b in Ashok et al., 2007). Unlike a canonical ENSO, there is a substantial meridional asymmetry in the SST field such that there is warming in the NH and cooling in the SH in EOF 3 as well as in the solar signal. Note that the solar signal has greater spatial extent, from the Pacific to Atlantic sectors, while that of EOF3 is confined mainly to the Pacific sector.

Cold tongues in tropical SSTs develop during boreal summer due to the Asian monsoon circulation (Wang, 1994). Therefore, the solar influence in the tropics is investigated for this season. Figure 11 shows correlation coefficients for boreal summer (JJA) between the solar index and a) SSTs, c) meridional wind velocity at 925 hPa, and e) out-going longwave radiation (OLR). Summertime climatologies are also displayed below the respective correlation plots; Figure 11b depicts climatological SSTs (contours) and their deviation from the zonal mean SST (color shading). The climatological northward component of the wind velocity at 925 hPa is displayed with 2 m s$^{-1}$ contours (Fig. 11d). Figure 11e shows climatological OLR (color shading). Regions of negative solar SST signals (Fig. 11a) roughly coincide with regions of low climatological SST with respect to the zonal mean, such as in the southeastern Pacific, the South Atlantic, and the coastal Arabian Sea. These sectors are also characterized by strong cross-equatorial winds along the continents (Fig. 11d). During periods of high solar activity a consistent increase in northward wind occurs in these regions (Fig. 11c). The correlation coefficients between the solar index (F10.7) and the OLR do not show a uniform increase of convective activity in the monsoon regions (lower OLR regions in Fig. 11f). The convective activity around the equatorial NH (0–10°N), such as over the Indian Ocean, South America, and Africa, is suppressed, while the convective activity of the off-equatorial NH (15°N–20°N) in the Asian sector tends to be enhanced. Thus, the tropical solar influence is not characterized by a strengthening of the global monsoon circulation, but rather a northward shift of the convergence zone or the ascending branch of the Hadley circulation. This shift also introduces longitudinal structure in the SSTs due to the asymmetric distribution of the continents.

## 4 Stratosphere–troposphere dynamical coupling processes

The results of the observational analysis so far suggest that the surface solar signals in both the tropics and the extra-tropics are related to the stratosphere through changes in the stratospheric westerly jet. Because of strong and complex feedbacks inherent in the atmosphere–ocean system, it is not easy to understand from observations alone how stratospheric circulation changes globally affect the troposphere.

Therefore, we now compare the observed surface solar signals with the response obtained from an idealized coupled atmosphere–ocean model experiment. In this experiment (Yukimoto and Kodera, 2007), stratospheric zonal winds are forced by the addition of zonal angular momentum in the winter stratosphere at levels above 100 hPa in the Meteorological Research Institute (MRI) coupled atmosphere–ocean general circulation model (MRI-CGCM2.3) (Yukimoto et al., 2006). The momentum forcing is essentially the same as that used by Thuburn and Craig (2000) except that the momentum forcing ($F_m$) is applied only in the winter hemisphere with seasonal variations, as follows.

$$F_m = A_0 f(p)(\sin 2\varphi)^2 \text{ MAX}\{0, \cos [2\pi(n-n_0) /365]\} \tag{2}$$

with $A_0$: maximum amplitude (5 m s$^{-1}$/day), $n$: day of the year, and $n_0$: central day of the winter (15 Jan in the NH, and 15 Jul in the SH). The vertical profile $f(p)$ is expressed as,

$$
\begin{aligned}
f(p) \ &= \ 1 & p &< 10 \\
&= \ \ln (p/100)/ \ln (0.1) & 10 &< p < 100 \\
&= 0 & p &> 100 \ ,
\end{aligned}
\tag{3}
$$

where $p$ denotes the pressure (hPa) and $\varphi$ denotes the latitude (radian).

Figure 12 shows the differences between the eastward and westward momentum (or strong and weak stratospheric westerly jet) experiments. Left- and right-hand panels are for July and January means of the last 50 years of a 100-year integration. Strong or weak stratospheric westerly jet in winter of extended period may exaggerate long-term feedback from the ocean, but we can see the effect of ocean more conspicuously. The momentum forcing and zonal mean zonal wind responses are shown in Fig. 12a and 12b. Although the momentum forcings are centered on 45° latitude in both hemispheres, the response in zonal mean zonal winds differs in austral and boreal winters. A strengthening of the polar-night jet occurs in January, approximately poleward of 30°N in the NH, and zonal mean zonal winds in the NH tropics decrease (noted by 'E' at the top of Fig. 12b). The deceleration occurs despite additional acceleration from the momentum forcing, due to the interaction with planetary waves. In contrast, in the SH in July, westerly winds weaken in the polar region.

Because stronger stratospheric westerly winds extend farther to lower latitudes in austral winter, a suppression of the ascending motion occurs more strongly in July in the tropics (Fig. 12c). As a consequence, stronger warming occurs around the tropical tropopause regions in July (Fig. 12d). Previous model studies (Thuburn and Craig, 2000; Kodera et al., 2011) showed that changes in stratospheric meridional circulation affect tropical convective activity through changes in static stability in the tropical tropopause region (Eguchi et al., 2015). In the present experiments also, suppression of equatorial ascending motion occurs in the troposphere in connection with the reduction of stratospheric mean meridional circulation change, as can be seen in the residual circulation differences in Fig. 12c. The extension of extra-tropical zonal mean zonal wind anomalies from the stratosphere to the troposphere occurs in association with a change in tropospheric wave activity as indicated by the Eliassen–Palm (E–P) flux (Fig. 12e). Upward-propagating waves are deflected equatorward around the tropopause region (300 hPa) and produce easterly zonal mean zonal wind anomalies in the subtropics, forming a pair of easterly and westerly zonal mean zonal wind anomalies at higher latitudes (Fig. 12b). This anomalous zonal mean zonal wind pattern also creates anomalous tropospheric warming around 40°N–45°N through the thermal wind balance (Fig. 12d). A particularly interesting response is found in the summer troposphere. Although no external forcing is applied in the summer hemisphere, an anomalous mid-latitude warming and wave activity persist in the troposphere, in particular in the SH. In fact, this latitudinal zone corresponds to the ocean frontal zone.

Figure 13a shows the horizontal structure of the annual mean SST differences between the stronger and weaker stratospheric westerly jet experiments, as in Fig. 12. This figure can be compared with the differences between high and low solar activity. The color shading in Fig. 13a shows the difference normalized by the standard deviation. Anomalous SST warming occurs around 40° latitude in both hemispheres, similar to the mid-latitude warming from the observations in response to the solar cycle in Fig. 1b. Cooling is also found in the equatorial southeastern Pacific and the Atlantic along the west coast of Africa, although it is quite small in the latter region. Note that the small response in the tropical Atlantic may be attributed to model deficiencies in low-level cloud formation. The cooling can be attributed to an increase of the cross-equatorial flow due to a suppression of rainfall near the equator, but an increase in off-equatorial regions (Fig. 13b). Cooling also appears in the coastal Arabian Sea in July (Fig. 13c) in connection with a strong northward meridional wind induced by an intensified Indian continent monsoon (Kodera, 2004). The increased convergence around the Indian continent is consistent with warming in the Bay of Bengal. These characteristics of the surface response to stratospheric westerly zonal wind changes are qualitatively consistent with the global surface solar signals from observations (Fig. 1).

**5 Centennial scale variation**

It is generally believed that changes in the solar UV produce regional effects in the troposphere, but have little impact on global mean temperatures (e.g., IPCC, 2013). However, this is not completely true for centennial solar variations. The effect of long-lasting weaker stratospheric polar vortices on tropospheric climate can be seen in the numerical experiment presented above.

Figure 14 shows annual mean surface air temperature differences between weak and strong stratospheric westerly polar-night jet experiments averaged over the last 50 years, as in Fig. 13. Note that the results of this experiment are more comparable with an extended period of extreme solar minimum (Maunder Minimum-like) conditions. The Earth's surface cools down remarkably. Global mean temperature decreases by about 0.5 K, although total solar irradiances are unchanged. This result arises because a weakening of the stratospheric polar vortex induces more frequent cold surges, which result in a larger snow cover extent in mid-latitudes. As a consequence, the Earth's albedo increases and the radiative balance changes without change in the TSI. The spatial structure of the temperature anomaly thus obtained is quite similar to that estimated from proxy data (see Fig. 3 in Shindell et al., 2001): a cooling over eastern Canada, eastern Europe to Russia, and northeast Asia, as well as a warming over the west coast of North America, west of Greenland, and Kamchatka, although the warming of the Middle East is shifted a little southwestward. This very good agreement of the global spatial structure of the surface temperature changes suggests a dynamical origin of the cooling during the late Maunder Minimum period. This is consistent with the conclusion of Mann et al. (2009), that the temperature variations of the Little Ice Age and the Medieval Climate Anomaly are of dynamical origin. Thus, centennial-scale solar signals could also be explained by a change in the spectral distribution of solar irradiance, with changes only in the UV part of the solar spectrum, even if the change in total energy was negligibly small.

## 6 Discussion and summary

### 6.1. Tropospheric processes

Annual mean surface temperature variation related to the solar cycle have been studied by several authors using long historical data as indicated in the introduction. In spite of different methods of analysis, the results present many similarities. The studies, however, focused on specific regions. Lohmann et al. (2004) indicate that significant solar signals are located in the Pacific sector, while Lean and Rind (2008) notice zonal warming signal along around 40° latitudes in both hemispheres. ZT2010 describe solar signal in each oceanic basin: in tropical Atlantic, the SST anomalies are cold south of the Equator during periods of high solar activity, but warm directly north of it. The northwestern Atlantic SST anomalies are cold, while those of the northwestern Pacific are warm except off the west coast of North America. The tropical eastern Pacific SST anomalies are cold, with exception of a thin warming band over the equatorial Pacific. The Indian Ocean shows warm anomalies. Gray et al. (2013) focus on lagged solar signal in the North Atlantic where NAO-like signal maximizes with a delay of about 3-years. They also report simultaneous solar signal such as weak cooling in the equatorial eastern Pacific, warming in northwest Pacific and a band of weak warming around 50°S–60°S. These solar signals reported by previous studies and describe above can be found in Fig. 1, except for the warming around 50°S –60°S. In fact, although less significant, larger warmings can also be recognized in the result of Gray et al. (2013) around 40°S latitude west of South America and Australia similar to the present study and others. As for the origin of these surface solar signals, Lean and Rind (2008) consider an involvement of the large-scale dynamical circulation of the atmosphere. ZT2010 consider dynamical response involving ocean-atmosphere couplings over Pacific and Atlantic Oceans, but they attribute direct solar heating effect for uniform warming of small scale Indian Ocean.

In these studies, no consideration is made, however, about concrete atmospheric processes which may induce solar signals in different regions.

In order to understand the physical processes, we first characterized positive and negative solar signals distributed over the globe. The mid-latitudes warming during high solar activity periods in northwestern Pacific and that of Southern Ocean around 40° latitudes are, in fact, located in the oceanic frontal zone as illustrated in Fig. 1. Negative SST anomalies in the tropical SH eastern Pacific and the Atlantic, are formed in the cold tongue regions. Therefore, we may be able to group the solar signals in surface temperature as follows:

1. Mid-latitudes warming around the ocean frontal zones.
2. Cooling in the tropical cold tongue regions.

In addition, warming over the sub-polar Eurasian continent and cooling in the west of Greenland can be attributed as solar signals related to the planetary wave structure (Fig. 3). Warming is also found over Antarctica with reanalysis data, which needs to be verified with more direct observational data. The observed feature in the SSTs in Fig. 1 is well reproduced in the result of numerical experiment in Fig. 13, which is a difference between two 50-years means of strong and weak stratospheric westerly jet conditions. The mid-latitudes warming in the northwestern Atlantic sector in the model appears together with the north-western Pacific sector because of the sufficient time for the response.

**6.1.1. Oceanic frontal zone**

Tropospheric part of Figs. 5, 6 and 7 are combined in Fig. 15 to illustrate the process suggested to produce tropospheric solar signal. Tropospheric temperature signal is associated with a downward penetration of zonal wind anomalies. Note that the statistically significance is higher in the zonal wind field than that of temperature (Fig. 15). A pair of warming and cooling is formed at both sides of the axis of the zonal mean zonal wind anomaly consistent with the thermal wind relationship. Westerly jet and ocean interaction is especially strong around the baroclinic zone where meridional temperature gradient is large. Numerical experiment of Nakamura et al. (2008) suggests an important role of oceanic frontal zones in creating variability in the tropospheric westerly winds through modifications of baroclinic waves. Impact of the momentum forcing experiment of stratospheric westerly jet also produces warmings around the baroclinic zone in the NH as well as in the SH (Fig. 13): Anomalous mid-latitude warming and wave activity persist in the troposphere, during the summer, although no external forcing is applied. Thus, anomalous temperatures in the baroclinic zone are maintained throughout the year, which could be the reason for significant and persistent solar signal in the annual mean temperature field. A possible role of ocean feedback to enhance stratospheric impact is also discussed in Yukimoto and Kodera (2007) and Misios and Schmidt (2013).

As reported by ZT2010, solar signal in mid-latitudes of North Pacific and North Atlantic are different. Cooling anomalies in the Atlantic sector are replaced by warm anomalies with a delay of about 3 years (Gray et al., 2013; Andrews et al. 2015). Scaife et al. (2013) demonstrated the role of ocean heat content producing this delayed effect. However, their calculated delays from a mechanistic model are too small to explain the observed signals. In the following, we suggest an additional mechanism which may lead to the delayed response in the Atlantic sector. During high solar activity, anomalous warmings develop over both Pacific and Atlantic sectors. However, the latitudinal position of these warming anomalies are different between the Pacific and Atlantic sectors due to the structure of the stationary planetary waves: in the Pacific, warming occurs around the frontal region, whereas that in Atlantic occurs in the subtropics, i.e. far south of the frontal zone as indicated in Fig. 8b. Warm anomalies in Atlantic sector subsequently shift northward to reach the frontal zone in 2 to 3 years (Fig. 9). The development of NAO-like solar signal could be expected around ocean frontal zone through positive feedback between the baroclinic waves and SSTs (see Supplementary Fig. 2 in Thiéblemont et al., 2015). In this respect, it is important to differentiate the spatial structures of the NAO and AO (Kodera and Kuroda, 2004; Wang et al., 2005): NAO is a regional meridional dipole pattern over North Atlantic due to interaction between the zonal wind and transient eddies, whereas the AO involves changes in planetary waves and polar vortex. The transformation of solar signal from AO-like to NAO-like in 3-years, such as simulated in a coupled ocean chemistry climate model experiment (Supplementary Fig. 3 of Thiéblemont et al., 2015), could be interpreted this way.

### 6.1.2. Cold tongue region

Van Loon et al. (2007) and Meehl et al. (2008, 2009) suggested that the tropospheric solar influence originates from an amplification by atmosphere–ocean interaction in the tropical Pacific; i.e., a modulation of the ENSO cycle. However, cooling anomalies during high solar activity does not solely occur in the Pacific tropical cold tongue regions, but in the Altantic as well. ZT2010 note also that the cooling in the Atlantic sector of the tropical SH is accompanied by a warming northward of the equator. The tropical Atlantic Ocean has no self-sustaining oscillation mode, unlike the tropical Pacific, but it can respond to external forcing with a north–south SST seesaw through the interaction of wind, evaporation, and SST (Xie and Tanimoto, 1998). Such a dipole pattern is discernible in Fig. 10. In fact, variations with the solar cycle of tropical Atlantic SSTs associated with cross-equatorial meridional winds have already been reported (Lim et al., 2006; Suh and Lim, 2006). Thus, the solar signal characterized by a simultaneous cooling in the two cold tongue regions could be understood from a northward shift of the convergence zone (i.e., the ascending branch of the Hadley cell) during boreal summer. Stronger southeasterly winds produce cooling in the equatorial SH west of the continents. These anomalies develop and are maintained through wind–evaporation–SST (WES) feedback, similar to that which creates a northward-displaced inter tropical convergence zone (ITCZ) in the climatological state (Xie, 2004).

Figure 16 summarizes troposphere atmospheric processes associated with the prominent solar signals on the Earth's surface: i) changes in westerlies associated with anomalous warming equatorward of the central location of anomalous westerlies and

ii) northward shift of the tropical convergence zone associated with cooling west of the South American and African continents through modulation of cross-equatorial flow.

## 6.2. Connection with the stratosphere

The question now is, how can the tropospheric circulation changes as illustrated in Fig. 16 be associated to the solar cycle. It is evident that decreased cloud coverage due to fewer cosmic rays during high solar activity cannot explain such a cooling over the cold tongue regions, where low-level clouds usually form (Kristjánsson et al., 2004). In the case of direct solar impact on the Earth's surface through visible light, one should expect a warming of the tropics similar to that observed in climatological SST distribution in Fig. 2e. For this, it needs to explain why warmings related with the 11-year solar cycle appear rather in
mid-latitudes (Figs. 2d and 5a).

We suggest that several aspects of the solar signal on the Earth's surface describe in our study may be explained by solar UV heating changes in the upper stratosphere which penetrate to the troposphere through two pathways: the stratospheric westerly jet in the extra-tropics, and the stratospheric mean meridional circulation in the tropics, as suggested by Kodera and Kuroda
(2002). The mid-latitude warming on the Earth's surface through the solar signal can be understood as produced in association with the downward penetration of zonal mean wind anomalies from the upper stratosphere during winter to spring in both hemispheres (Fig. 6). Connection in the tropics through changes in vertical velocity is suggested to occur through change in tropical lower stratospheric temperature (see Fig. 1 of Kodera and Shibata, 2006). Tropical lower stratospheric temperature change associated with the solar cycle is larger in boreal summer (Jul-August) than that in austral summer (November-
December) as can be seen in Fig. 7 of Labitzke (2001). Tropical cooling associated with the solar cycle develops during summer to autumn following the increase of cross-equatorial flow in boreal summer. It is difficult to investigate how the lower stratospheric temperature affects the tropical convective activity for solar cycle scale variation based on observational data only, however. In this respect, the results of our momentum experiments suggest that the change in the BD circulation in the stratosphere can be connected to the raising branch of the Hadley circulation and modulate upward velocity in the tropics (Fig.
11).

There are some differences in solar signals between the NH and SH both in the stratosphere and troposphere, which also needs to be explained. To emphasize the initial role of the solar UV heating in the upper stratosphere, only the early winter situation was shown in Figure 15 of Kodera and Kuroda (2002). However, the stratospheric circulation evolves seasonally from a
30 radiatively controlled to a dynamically controlled state. Here, we show these two stages schematically in Fig. 17 based on previous studies (Kodera and Kuroda, 2002; Matthes et al., 2006; Matthes et al., 2013). Increased solar UV heating in the tropics produces only a small increase in the subtropical jet in the case of no interaction with waves (Fig. 17a). However, such a small initial effect can be amplified through wave–mean flow interactions. During early winter, when planetary wave forcing is small, the waves (green arrows) are deflected at the stratopause subtropical jet (Fig. 17b). In this case, the downward

extension of the subtropical jet occurs in association with significant tropical warming and mid-latitude cooling (Fig. 17b) as shown in Kodera and Kuroda (2002). In contrast, when planetary wave forcing becomes large enough in late winter to spring, the waves penetrate the subtropical upper stratosphere–stratopause region leading to a poleward shift of the westerly jet (Dunkerton, 2000) (Fig. 17c). Enhanced vertical wave propagation along the polar-night jet results in an increased convergence of waves in the upper stratosphere, on the one hand, while on the other hand it induces divergence in the lower stratosphere, by which westerly anomalies descend into the polar region (Kuroda and Kodera, 1999). This results in a warming in the polar region of the upper stratosphere, but a cooling (or a reduction of the warming) in the tropical stratosphere due to an enhanced mean meridional circulation, as schematically illustrated in Fig. 17c. Thus, the differences in the solar signal characteristics between the SH and the NH can be understood by the different durations of the radiatively and dynamically controlled stages related to different planetary wave activity.

Lower stratospheric tropical heating was proposed as possible origin of the solar influence on the troposphere (Haigh et al., 2005; Simpson et al., 2009). The solar signal in the NH is, in fact, transmitted from the upper stratosphere to the surface through a poleward–downward shift of anomalous zonal mean zonal wind, which creates a NAM-like structure in the troposphere. In the SH the planetary wave forcing is smaller, meaning that, climatologically, the radiatively controlled stage lasts longer than in the NH. As a consequence, the stratopause subtropical jet develops and extends to lower levels without a large poleward shift, meaning in turn that tropospheric solar signals in the SH do not resemble the SAM which is sensitive to the variability in the westerly jet in high latitude. On the other hand, change in westerlies in lower latitudes produces larger solar signals in the tropical lower stratosphere.

The dynamical solar influence from the stratosphere can be reproduced by forcing stratospheric zonal mean winds in a coupled atmosphere–ocean general circulation model as shown in Figs. 11 and 12. A realistic numerical experiment with solar UV forcing in a general circulation model without an interactive ocean successfully reproduced the downward propagation of solar signals during NH winter (e.g., Matthes et al., 2006). More recent advanced middle atmosphere chemistry-climate models including the feedback to the ocean, can now simulate zonal mean zonal wind variations with the solar cycle and their extension to the troposphere in both hemispheres as well as the observed differences in the NH and the SH (see e.g. Figs. 10 and 11 of Hood et al., 2015). The fact that the realistic solar response is obtained only from the models capable to reproduce realistic upper stratospheric ozone variability, also supports the downward penetration of the solar influence from the upper stratosphere.

It should finally be noted that centennial circulation changes produced in the stratosphere can affect global mean surface temperature through changes in the Earth's surface condition without changes in total solar irradiance. The following processes, however, need further clarification: i) the role of ocean fronts and atmospheric baroclinic eddies in the downward extension of zonal mean zonal winds from the stratosphere, and ii) the role of tropical convection in interactions between the stratospheric

mean meridional circulation and the Hadley circulation. Concerning the La Niña-like SST anomaly, Roy and Haigh (2012) confirmed a tendency for La Niña to occur more frequently during the peak year of the solar cycle as previously suggested by van Loon et al. (2007). However, their peak year is only one year among 11-years of a solar cycle. The SST pattern related to the entire solar cycle extracted by a CMD or MLR methods rather resembles to El Niño Modoki of Ashok et al. (2007) as illustrated in Fig. 10. In fact, La Niña like pattern at the solar peak year rapidly evolves to a different pattern in one or two years (Meehl and Arblaster, 2009). Such non-linear aspect of the interaction between the ENSO and solar cycle will be addressed in a separate study.

## Acknowledgements

This work was supported in part by JSPS Grants-in-Aid for Scientific Research (S)24224011 and (C)25340010. KM and RT acknowledge support from the Helmholtz-University Young Investigators Group NATHAN funded by the Helmholtz-Association through the President's Initiative and Networking Fund and the GEOMAR Helmholtz Centre for Ocean Research Kiel. This work has been conducted within the frame of the WCRP/SPARC SOLARIS-HEPPA activity.

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

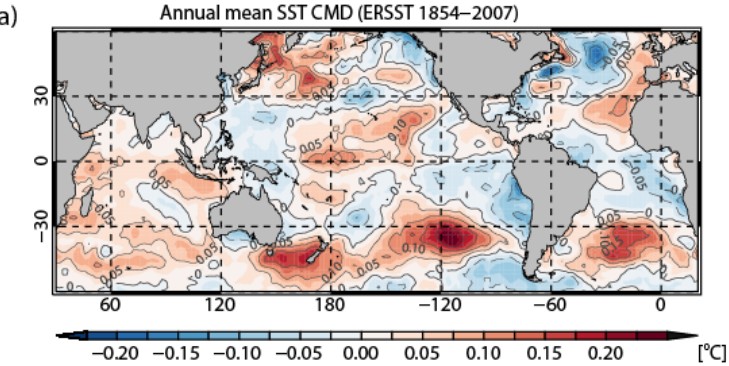

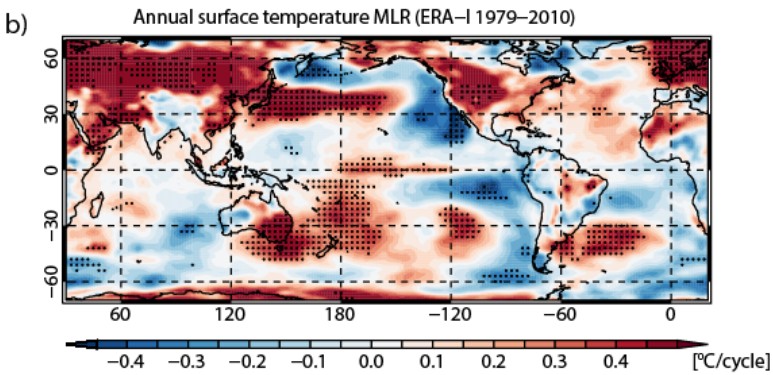

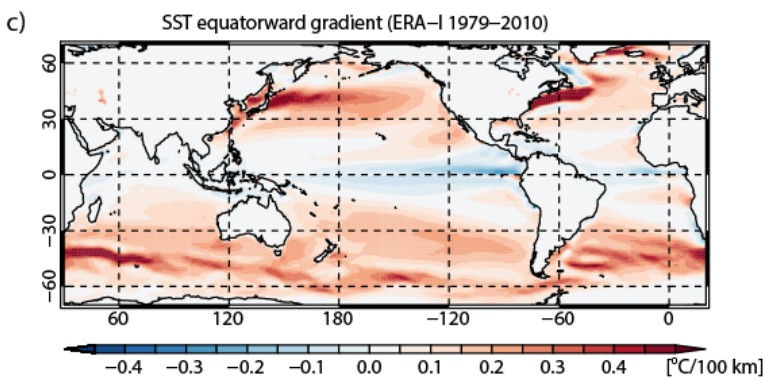

**Figure 1: T a) Annual mean SST anomaly extracted by the same CMD analysis as in Zhou and Tung (2010) for the period 1854-2007. b) Annual solar index regression coefficient of the surface temperature derived by applying the MLR model to ERA-I data for the period 1979–2010. Stippled areas indicate statistical significance at the 95% level. c) Equatorward gradient of annual mean climatological SST.**

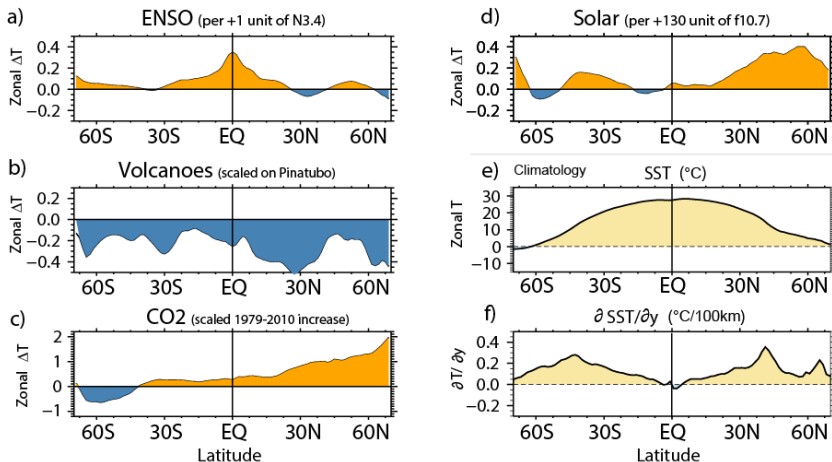

**Figure 2: MLR analysis of the annual zonal mean surface temperature from ERA-I, calculated for the period 1979–2010, for (a) ENSO, (b) volcanic activity, and (c) CO₂ concentration, and (d) solar activity. Climatological zonal mean SSTs and their equatorward meridional gradient are also shown in (e) and (f), respectively.**

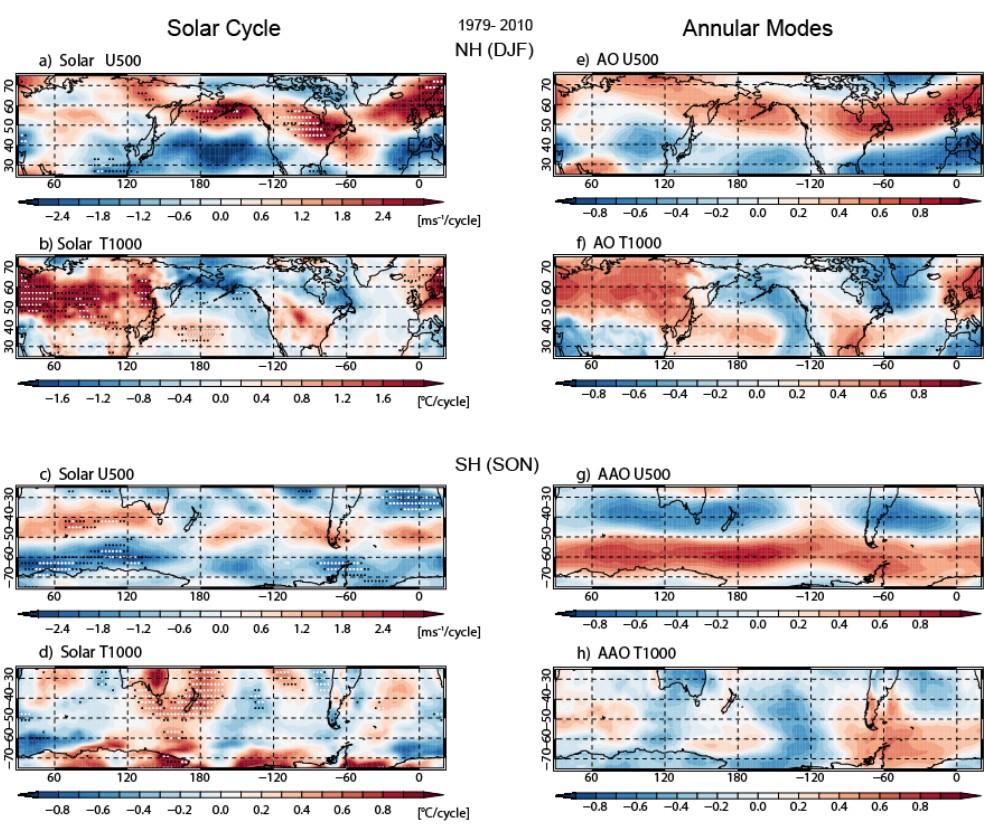

**Figure 3: Solar regression coefficient extracted by the MLR technique for the DJF mean NH (a) 500 hPa zonal mean wind, and (b) surface temperature. (c and d) Same as (a and b), but for the SON mean in the SH. (e and f) Same as (a and b), except for the correlation with surface NAM index. (g and h): same as (c and d), except for the surface SAM index. The period of analysis is 1979–2010. Stippled regions with black and white dots indicate statistical significance at the 90% and 95% level, respectively.**

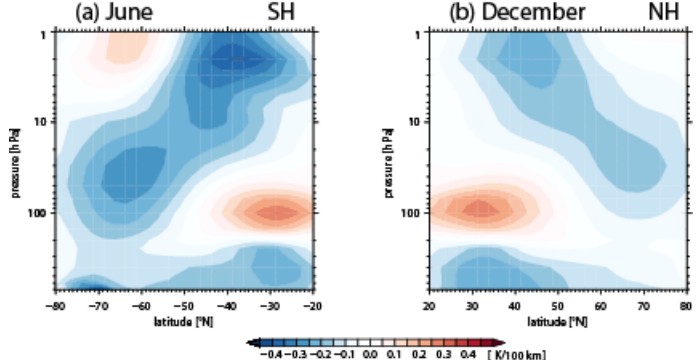

**Figure 4: Meridional sections of the climatological poleward temperature gradient around the winter solstice: (a) SH June, (b) NH December.**

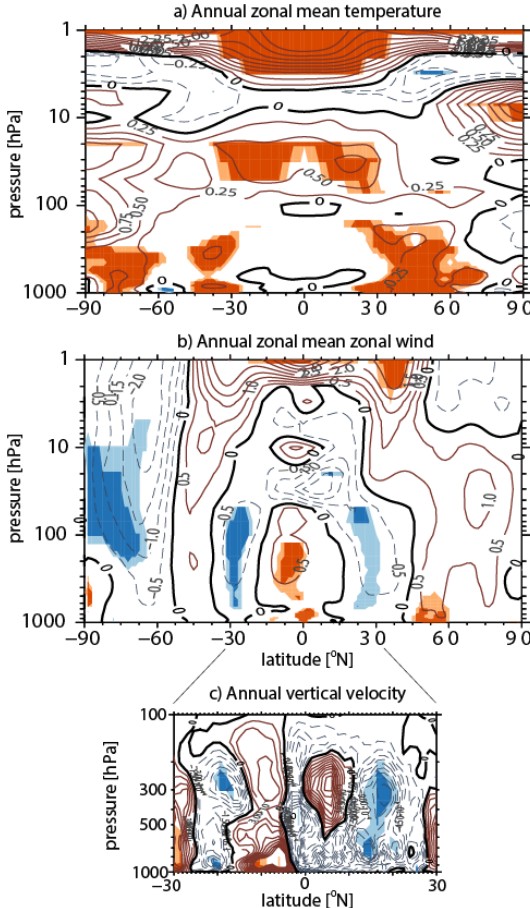

Figure 5: Solar regression coefficients of the annual-zonal mean a) air temperature, b) zonal wind, and c) vertical velocity in the tropical troposphere. Solid (dashed) contours indicate positive (negative) values and are drawn every (a) 0.25 K, (b) 0.5 m s⁻¹, and (c) 5 m /day. Areas of 90% and 95% statistical significance are shown by light and dark shading, respectively, in red (positive) and blue (negative).

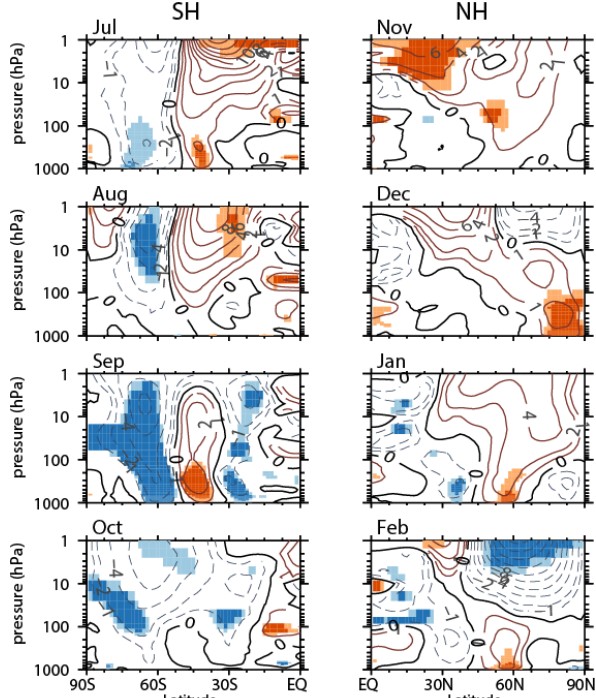

**Figure 6: Monthly solar regression coefficient of zonal-mean zonal winds in (left) July, August, September, and October in the SH, and (right) November, December, January, and February in the NH. Solid (dashed) contours indicate positive (negative) values and are drawn every 1 m s⁻¹. Areas of 90% and 95% statistical significance are shown by light and dark shading, respectively, in red (positive) and blue (negative).**

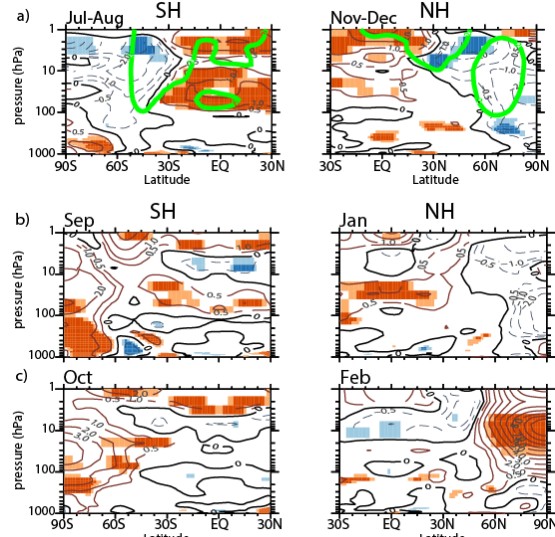

**Figure 7: (a) Same as Fig. 6, except for 2-month mean air temperature, July–August in the SH (left), and November–December in the NH (right). Green lines indicate 2 m s⁻¹ contours of the corresponding zonal mean zonal wind. (b) Same as (a), except for monthly mean temperature in September (left) and January (right). (c) Same as (b), except for October (left) and February (right). Contour interval for temperature is 0.5K.**

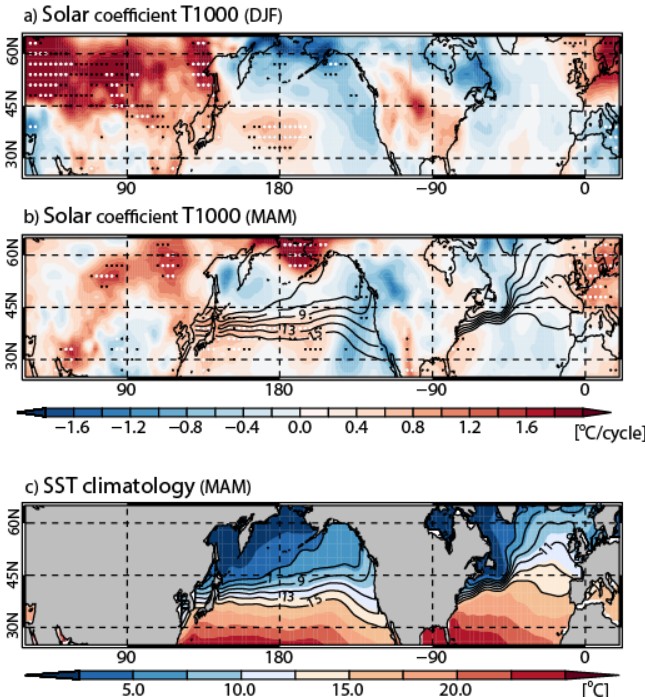

**Figure 8: (a and b) Solar regression coefficient of the surface temperature (at 1000 hPa) over the NH mid-latitudes for (a) DJF and (b) MAM. Isothermal lines over oceans in (b) represent climatological SSTs displayed in the bottom panel. c) Climatological mean SST in spring (MAM). Stippled areas with black and white dots indicate statistical significance at the 90%, and 95% levels, respectively.**

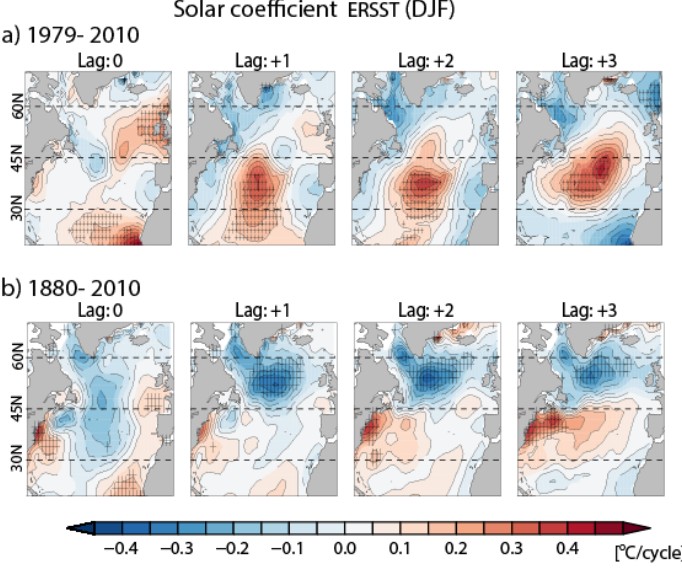

**Figure 9: Solar regression coefficients of winter mean SST in the North Atlantic sector extracted from ERSST at lag times of 0, 1, 2, and 3 years. for (a) 1979 to 2010, and (b) 1880 to 2010. Stippled areas indicate statistical significance at the 90% level.**

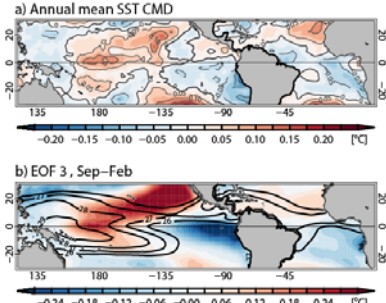

**Figure 10: (a) Same as Fig. 1a, except for the tropical Pacific and Atlantic sectors only (30°S−30°N, 120°E–20°E). (b) SST spatial structure of the third EOF in September–February for the period 1890–2012 (Colour shading). Contours indicate climatological SST.**

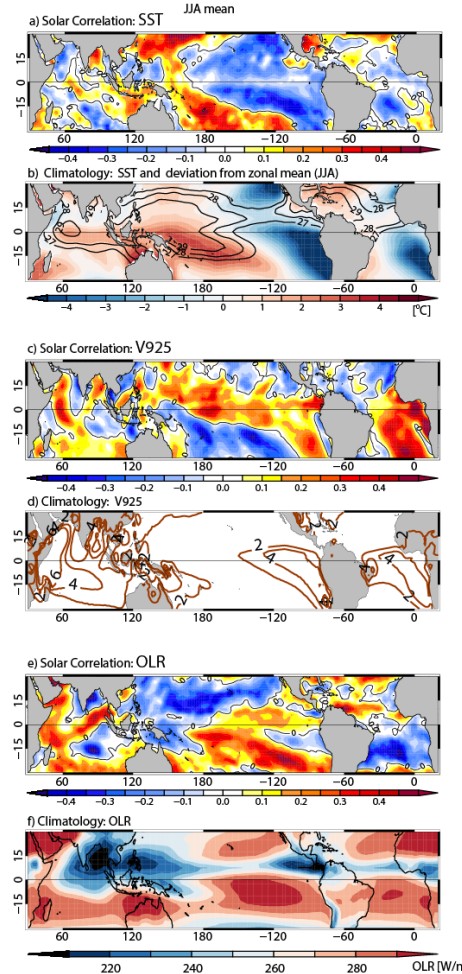

**Figure 11: Boreal summer (JJA) solar signal in (a) SST, (c) meridional winds at 925 hPa, and (e) OLR, presented as correlations with the solar index for the period 1979–2010. b) JJA mean climatological SST, with contours for 27°, 28°, and 29°C, and color shading denoting the deviation from the latitudinal mean SST. d) Climatological JJA northward wind component at 925 hPa (contours every 2 m s⁻¹). f) Climatological JJA OLR (color shading).**

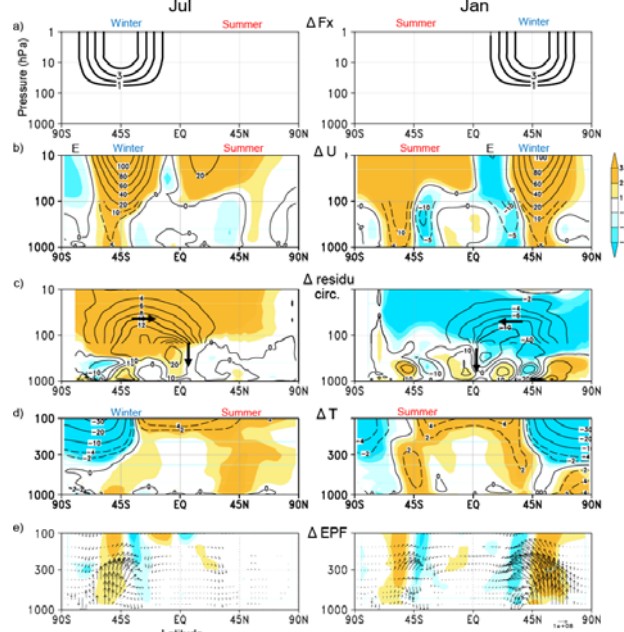

**Figure 12: Difference between strong and weak stratospheric westerly jet experiments by Yukimoto and Kodera (2007) in July (left) and January (right): (a) Zonal momentum forcing (m s⁻¹/day), (b) zonal-mean zonal winds (m s⁻¹), (c) mean meridional residual circulation (10⁹ kg s⁻¹), (d) zonal-mean air temperature (K), and (e) E–P fluxes (m² s⁻²) (arrows) and their divergence (color shading). Color shading indicates differences normalized by the standard deviation.**

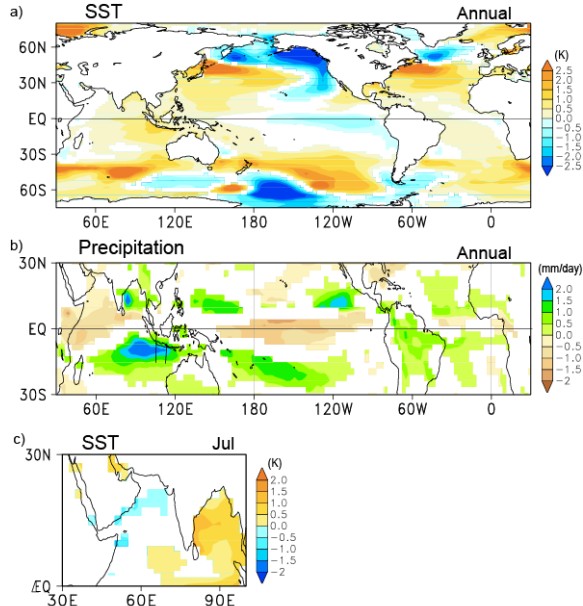

**Figure 13: a) Differences in the annual mean SST between strong and weak stratospheric westerly jet experiments, similar to Fig. 11. b) Same as (a), except for the annual mean precipitation. c) Same as (a), except for July mean SST in the Indian Ocean sector. Units are (a) K, (b) mm/day, and (c) K. Color shading indicates regions where statistical significance exceeds the 95% level.**

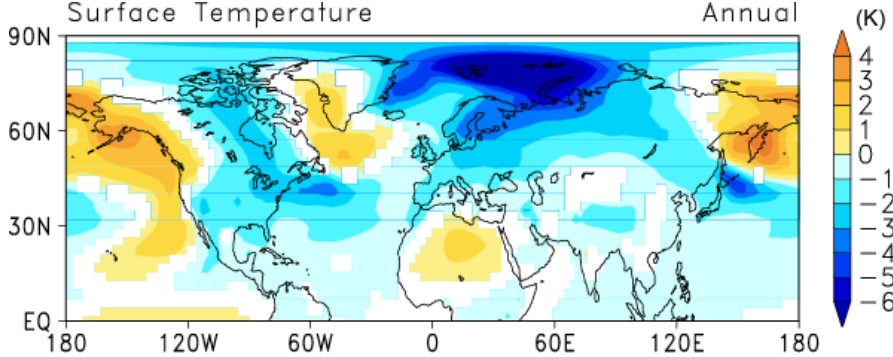

**Figure 14: Similar to Fig. 13, except for annual mean surface temperature differences between weak and strong stratospheric westerly jet experiments, comparable to an extended period of extreme solar minimum (Maunder Minimum-like) conditions. Color shading indicates regions where statistical significance exceeds the 95% level.**

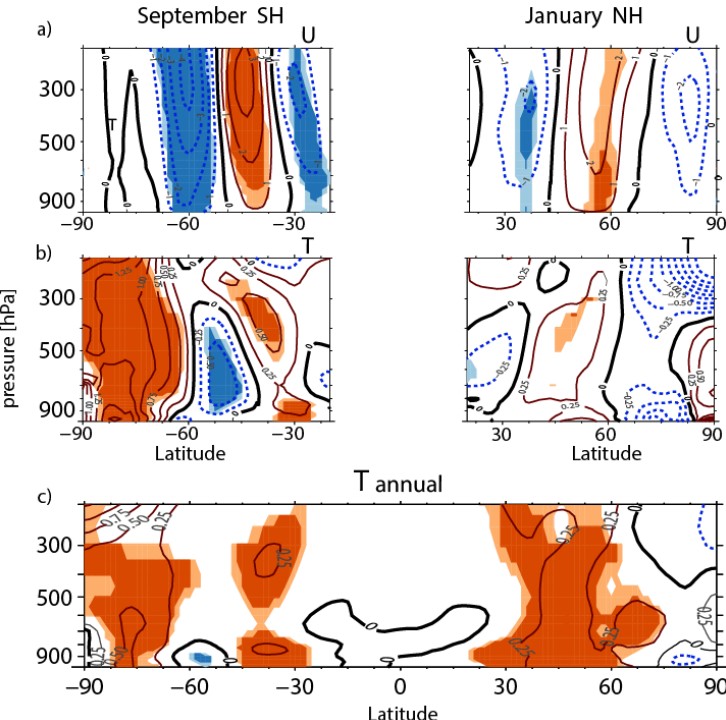

**Figure 15: (a) Same monthly solar regression coefficient of zonal-mean zonal winds as in Fig. 6, except for the tropospheric part: (left) SH in September, and (right) NH in January. (b) Same as in (a), except for the temperature signal in Fig. 7. (c) Tropospheric part of solar regression coefficients of the annual-zonal mean air temperature in Fig. 5a. Contour intervals are 0.25K for temperature and 1 ms$^{-1}$ for wind.**

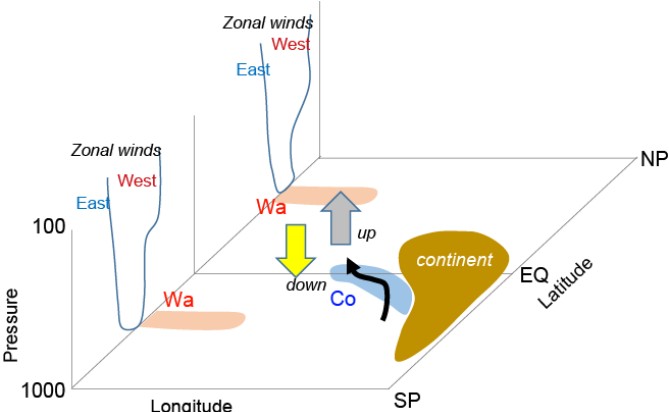

**Figure 16: Schematic presentation of tropospheric processes related to the solar signals in the Earth's surface temperature: i) changes in westerly jet (West or East) produces anomalous warming (Wa) equatorward of the centre of anomalous westerlies. ii) Northward shift of the tropical convergence zone (up and down) produces coolings (Co) west of continent through modulation of cross-equatorial flow (arrow).**

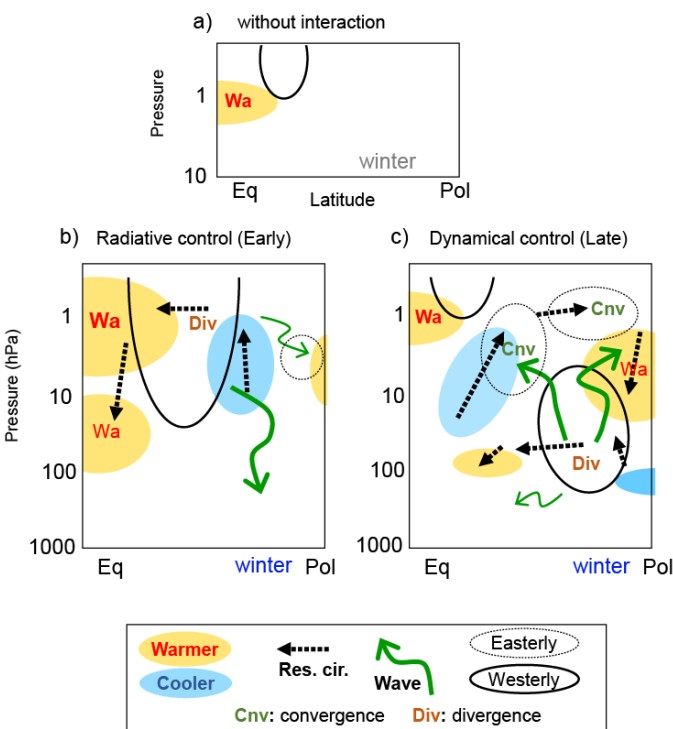

**Figure 17: Schematic presentation of the solar influence on the winter stratosphere. (a) Hypothetical response to solar UV heating without interaction with planetary waves. (b) Early winter when solar radiative forcing dominates, and (c) late winter when dynamical forcing from the troposphere becomes more important. See text for details.**

