# Peer review of "How can we understand the global distribution of the solar cycle signal on the Earth's surface?"

_Atmospheric Chemistry and Physics, 2016_

## Referee Comment (RC1) · Anonymous Referee #1 · 5 Apr 2016

Summary

The authors perform a variety of methods to understand the atmospheric response and surface response to solar changes. They mix observations, reanalysis and model experiments. The study culminates in the modelling assessment, where induced momentum (analogous to solar forcing) is directed into experiments to assess the surface response.

The manuscript is well written, and allows the reader plenty of spin up time in on solar influence on climate. The methods seem sound, although more discussion over different methods would help guide follow on research from this study.

My main criticism is that the paper has a lot of repeated material already in the literature. In fact up to figure 8 there is nothing substantially new. I think the most exciting

parts of the paper certainly come from the model experiments, which culminate in figures 10 and 11. If I were the authors, I would have expanded this section more (at the expense of the early sections), or simply submitted it is a shorter letter based article. Nevertheless, as the authors have submitted the paper as it is, it may be OK to publish, although I believe it needs to be made far more clear that Figures 1-8 are mainly reproducing previous work. Therefore the manuscript needs to be better referenced, and clear in the discussion of the paper structure.

General Comments

1. Make the paper clearer as to what is new and what is not. 2. There are inherent issues with some of the analysis that the authors use. Namely that solar signals can interact in a nonlinear way with the atmosphere and even more so the surface. We know some of the responses are non-linear, and therefore multiple linear regression may not be the most appropriate tool. My feeling is the MLR is probably OK for some assessment of the surface, but the issues with it should certainly be addressed in a standalone paragraph. For instance, machine learning methods (Blume et al, 2012) which are naturally non-linear, optimal detection (Stott et al, 2003; Mitchell, 2015) which gets around some of the non-linearity by using model predicted responses as the regressors (they also do not assume noise free regressors, another issue with the standard MLR), and final non-linear attribution, (Kuchar et al, 2015). The latter study does some comparison with the MLR technique as well, although does not directly address your analysis. 3. From the title I thought it would be a rather different paper. I do not think you have answered the question 'how can we understand...', I think you have simply performed an analysis of the surface response. I would therefore change the title.

Minor comments

P1L27: This sentence does not fit so well. In longer term studies (of centuries) solar influence on climate has been known about for a while. Do you mean just short term?

P2L11: 'global mean temperature' to you mean 'surface temperature'?

Section 2.2: Here I would address my General Point 2.

P4L14: A number of studies use additional regressors. Maybe a point or two on why these are OK.

P5L4-5: Maybe cite some papers that look at solar influence on climate using the Tiao method.

P5L27-30: It is not clear to me exactly where you refer to here. Is it literature, or is it panels a and b? If the latter, I still do not see all the features that are mentioned.

P6L5 2-3 years should probably be 2-4 years. In the literature it is often written in both ways, but I think if you look at the figures in the relevant literature, the signals at 2 years are as large as those at 4 years (with the signal max at 3 years).

P6L11-22: Is there perhaps cross correlation in the regressors between solar and volcanic? For instance the response seems anticorrelated say at 45N which is a max in solar, and a min in volcanoes.

P6L32: A forcing of the vortex nearly always leads to a response in the NAM, so why is it remarkable?

P8L15-16: The temperature response seems very large over Eurasia. Is this real? I find it hard to believe that the temperature response is over 2K. I think this needs to be investigated and discussed more.

P10L15-22: Is there a QBO in the model? How does the momentum forcing interact with the QBO. Surely at some points they will not be consistent with the H-T relationship?

P11L8-19: The authors are very sure about the casual links here. I think they need to be more speculative about the comments, or back it up with modelling evidence from their model.

P12L13-19: So are the authors suggesting they do not believe the Haigh mechanism?

[Figure]

I think it is still important, but the paragraph does not read that way. I would also cite Simpson et al, 2009.

P13L13-16: Hood et al only show a subset of models, and not even all the coupled chemistry models from CMIP-5. Are there better (or additional) references that could expand on this point?

Figures

Figure 1: I would make a and b more comparable. Use the same contour intervals and only plot of the oceans. Also use the same latitude ranges.

Figure 4: There is a lot of detail in panel c, and it can't really be seen. Can you enlarge it to the size of the other panels.

References

Blume, Christian, Katja Matthes, and Illia Horenko. "Supervised learning approaches to classify sudden stratospheric warming events." Journal of the Atmospheric Sciences 69.6 (2012): 1824-1840.

Kuchar, Ales, et al. "The 11-year solar cycle in current reanalyses: a (non) linear attribution study of the middle atmosphere." Atmospheric Chemistry and Physics 15.12 (2015): 6879-6895.

Mitchell, D. M. "Attributing the forced components of observed stratospheric temperature variability to external drivers." Quarterly Journal of the Royal Meteorological Society (2015).

Simpson, Isla R., Michael Blackburn, and Joanna D. Haigh. "The role of eddies in driving the tropospheric response to stratospheric heating perturbations." Journal of the Atmospheric Sciences 66.5 (2009): 1347-1365

Stott, Peter A., Gareth S. Jones, and John FB Mitchell. "Do models underestimate the solar contribution to recent climate change?." Journal of Climate 16.24 (2003): 4079-

4093.
* * *

---

## Referee Comment (RC2) · Anonymous Referee #2 · 7 Apr 2016

This paper studies the 11-year solar cycle signal in Earth's surface using historical datasets and the surface evidence is further supported by the zonal mean vertical profile using ERA-Interim and previously archived model simulations. The authors present many surface and zonal mean quantities that are composited between high and low solar years. Although majority of the results presented here are known or previously published, this paper could still be informative because it provides an up-to-date and comprehensive summary of the atmospheric response to the 11-year solar cycle in the observational data sets.

The authors have attempted to examine the dynamical mechanism by which the 11 year solar cycle signal is transmitted from the tropical upper stratosphere to the surface. They suggest that the observed surface signals are largely resulted from circulation changes in the upper stratosphere through downward migration of zonal mean

anomalies and changes in the stratospheric mean meridional circulation. The authors' argument on this point is demonstrated mostly by using a model simulation where westward and eastward momentum forcing was applied to the entire column of the winter stratosphere polar vortex. The initial solar UV forcing however normally confines to the subtropical upper stratosphere, i.e. above 10 hPa. Thus, it differs significantly from the strong and weak polar vortex cases in their model simulation. Firstly, the solar UV effect at lower latitudes must be transmitted to middle to high latitude to produce a definite stronger vortex, which is not always easy in the real atmosphere. This is clearly demonstrated by the different responses in the SH and NH during winter sessions. The authors present no diagnostics of the wave-mean flow interaction or meridional circulation in the stratosphere based on observation or reanalysis data. Only if the wave forcing diagnostics from reanalysis data sets match those from their model simulations, the proposed mechanism can then be stated as the main mechanism for the solar signal seen in the SSTs or SATs. I therefore find that this part of the paper is not entirely convincing. The rather strong statement made by the authors about the cause and effect regarding the link between the surface signal and this mechanism should be tuned down and presents as one of the contributing mechanisms instead. If not, please provide additional supporting wave-mean flow interaction diagnostics using ERA-Interim or other reanalysis data sets.

The results are appropriate for ACP and the structure of the paper is sound. The clarity of paper may be improved by reducing the lengthy discussion. I have several specific comments that I would like to see addressed before the paper is published.

Major comments:

1. Lines 22-24. Abstract. As I stated previously, these statements are too strong given the momentum forcing applied in the model simulation differ largely from the actual solar UV forcing.

2. It appears to me that the atmospheric or tropospheric response in their model simulation (Figure 10) can only explain the early winter behaviour of the solar signal in the NH. It fails to explain the high latitude warming signature in the late NH winter and in SH spring and no signal in SH winter (Figure 6).

3. Lines 5-30, Page 4. MLR may be quite useful in studying a system in which the dependent variables are linearly related to the predictors in time. The assumption may hold for annual mean fields but will not be applicable for the seasons where non-linearity dominates. In NH winter, for instance, the authors have suggested that the stratospheric response to the 11-year solar UV cycle in early and late winter flips the sign. This suggests nonlinearity and may result in cancellation of solar signal there when a linear regression model is applied. It would be helpful to the readers if the authors make this point clearer.

4. From Figures 5 and 6, it is not clear to me how the surface temperature and circulation patterns are so-surely linked to the stratospheric circulation anomalies, as the way presented by the authors. In both hemispheres, little solar signals can be found in the polar temperature during middle winter (see Figure 6). In the NH, the mid-latitude troposphere and lower stratosphere show to be weakly warm in Nov-Dec, Jan and Feb while the polar region flips from cold to warming from Nov to Feb. Thus, why the upper level "causing" signals are effectively weaker than the "responding" signals near the surface? Or to what extent these winter temperature anomalies shown in Figure 6 contribute to the annual mean anomalies shown in Figures 1 and 4?

5. Line 31, section 3.4, page 8. Tropical solar signals appear to be important in this paper and the authors have devoted an entire subsection for it. However, in the abstract, it states "no warming in the tropics". Somehow, I feel that the authors need to provide the reason as why the tropic solar signals need to be specifically discussed given the most significant solar signals are found in the middle latitudes (See figure 1). Also, in what way the tropical solar signals are connected to the dynamical mechanism by which the 11 year solar cycle signal is transmitted from the tropical upper stratosphere to the surface?

6. Figure 12c is rather sudden and thus potentially confusing because the wave forcing and residual circulation anomalies in late winter are not supported by any of the analysis presented earlier in the manuscript based on either data or model simulations.

7. Lines 5-8, page 13. I cannot see the reason why a longer lasting radiatively controlled stage in the subtropical SH upper stratosphere can lead to an anomalous weakening of the stratospheric jet and warmer polar stratosphere (Figures 5 and 6). It appears to me that the argument based on dynamical versus radiative control is definitely valid in part but it remains not sufficient to explain all the stratospheric anomalies.

8. Lines 21-34, page 14. These sound much like results rather than discussion and concluding remarks. Suggest moving to an earlier section instead. As I have stated before, the composite difference estimated from the simulated weak and strong polar vortex are not exactly representative to actual solar UV forcing. First, the solar UV forcing has much smaller magnitude. Second, the solar UV effect is located much higher in altitude than the model simulation assumed. As a result, the solar UV effect should be much weaker than what has been suggested by the model simulation.

9. Some of the fields are quite messy (e.g. Figure 4b,c; Figure 6) or not statistical significance is shown (e.g. Figure 1a). Some of the features are not statistically significant but are discussed as the cause for the surface anomalies. I suggest that the discussion around these figures/features needs to be more careful.

Minor comments:

1. Line 11, abstract. "no warming in the tropics". This is not clear. "No warming" could imply either "cooling", "no signal" or "complex signal with longitudinal variation".

2. Line 14, abstract. "the subtropical jet". The term is not clear. The subtropical jet in the atmosphere often refers to the tropospheric subtropical jet. Here, the authors refer to the upper stratosphere subtropical jet. Climatologically speaking, there is no subtropical jet in the stratosphere anyway. There is only one jet in the stratosphere

which is the polar vortex which initializes at lower latitudes in early winter.

3. Line 1, page 2. "amplify" -> "act to amplify".

4. Line 13, page 2. Zhou and Tung (2010) not cited in the reference list.

5. Line 26-27, page 2. "Because solar signals in SLP data are inconsistent, probably due to the temporal and spatial limitations of the data, we instead study pressure or geopotential height fields ...". It is confusing firstly because the SLP is pressure, isn't it? Also, it is known that solar signal tends to wax and wane with the different periods under consideration. Would it be better that we admit that we still do not understand why it happens rather than blaming the data quality. The wax and wane can also be found in modern data sets such as ERA-40 or ERA-Interim.

6. Line 16, page 4. "predictorand" -> "predictors".

7. Line 19-20, page 7. "The differences in the latitudinal structure of the warming suggested ...". This is not clear especially from the annual mean field. These statement can only be said when other dynamically quantities are also analysed. Suggest to remove or cite references to support such claim.

---

## Author Comment (AC1) · 3 Jun 2016

Many thanks for your comments and suggestions on our manuscript. Here is our answers ( ==> ).

Anonymous Referee #1
Summary
The authors perform a variety of methods to understand the atmospheric response and surface response to solar changes. They mix observations, reanalysis and model experiments. The study culminates in the modelling assessment, where induced momentum (analogous to solar forcing) is directed into experiments to assess the surface response. The manuscript is well written, and allows the reader plenty of spin up time in on solar influence on climate. The methods seem sound, although more discussion over different methods would help guide follow on research from this study.

My main criticism is that the paper has a lot of repeated material already in the literature. In fact up to figure 8 there is nothing substantially new. I think the most exciting parts of the paper certainly come from the model experiments, which culminate in figures 10 and 11. If I were the authors, I would have expanded this section more (at the expense of the early sections), or simply submitted it is a shorter letter based article.

Nevertheless, as the authors have submitted the paper as it is, it may be OK to publish, although I believe it needs to be made far more clear that Figures 1-8 are mainly reproducing previous work. Therefore the manuscript needs to be better referenced, and clear in the discussion of the paper structure.

==> To make the point of the present study clearer, the title has been changed from "How can we understand the solar cycle signal ..." to "How can we understand the global distribution of the solar signal".

Similar solar signal in the surface temperature as in Fig. 1 can be found in previous papers. Many studies, however, focus on regional aspects. For instance, the paper by Meehl et al. (2008) illustrates only the Pacific sector, whereas that by Gray et al. (2013) focusing on the Atlantic sector, uses a map centered on the Greenwich meridian. In the case of Zhou and Tung (2010), having no interest in the spatial structure, a map according to a convention starting from 0° to 360° longitudes is used. If the results of the analysis were very similar, they are presented differently, in a way to what authors aim to study or demonstrate. In the present study, we investigate global features including a connected variation in the solar signal from the Pacific to the Atlantic sector. Therefore we need to produce figures appropriate for our study.

To indicate what is new in this paper, the following sentences were added. "It should be noted that most of the previous work investigated processes producing solar signals on the Earth's surface in a specific region. Little was done to understand the overall aspect of solar signals on the entire Earth's surface."

We also added references to previous similar MLR studies using meteorological reanalysis data: "The results of similar MLR analyses using meteorological reanalysis data have also been published (e.g., Frame and Gray, 2010; Chiodo et al., 2014; Mitchell et al., 2015a)."

**General Comments**

- Make the paper clearer as to what is new and what is not.

==> As mentioned above, most of the previous studies focus on regional aspects of solar influence if not a globally averaged temperature. What is new in this paper is a study of the processes which produce a global distribution of the solar signal in the surface temperature in the extratropics of the NH and SH, as well as in the tropics.

- 2. There are inherent issues with some of the analysis that the authors use. Namely that solar signals can interact in a nonlinear way with the atmosphere and even more so the surface. We know some of the responses are non-linear, and therefore multiple linear regression may not be the most appropriate tool. My feeling is the MLR is probably OK for some assessment of the surface, but the issues with it should certainly be addressed in a standalone paragraph. For instance, machine learning methods (Blume et al, 2012) which are naturally non-linear, optimal detection (Stott et al, 2003; Mitchell, 2015) which gets around some of the non-linearity by using model predicted responses as the regressors (they also do not assume noise free regressors, another issue with the standard MLR), and final non-linear attribution, (Kuchar et al, 2015). The latter study does some comparison with the MLR technique as well, although does not directly address your analysis.

==> We revised section 2.2 by adding the following paragraph to indicate the limitations of the MLR method as suggested by the reviewer:

"The use of the MLR approach to separate the contribution of different factors on climate variability has inherent limitations that should be kept in mind when analyzing the results. The MLR particularly relies on several assumptions that may not be valid in all cases. However, composite analysis of the monthly mean data based on two or three levels of solar activity (e.g., Kuroda and Kodera, 2002; Lu et al., 2011) also produces similar solar signals as those obtained from the MLR method (Figs. 5, 6). Therefore, in spite of the limitations, the MLR method may be useful to get approximate solar signals (Kuchar et al., 2015). We note that highly non-linear responses can be produced through the interaction between different forcings: for example between ENSO and solar signals (Marsh et al., 2007), solar and QBO signals (Matthes et al., 2013), as well as volcanic and solar signals (Chiodo et al., 2014). Usually this kind of interaction occurs at a specific location and time, which needs to be investigated in separate studies. Sophisticated attribution methods which can account for non-linearity have been used, such as machine learning methods (Blume et al, 2012), or optimal detection (Stott et al, 2003; Mitchell, 2016). Although these methods allow advanced statistics to be established, their limited interpretive capacities make it difficult to study physical mechanisms. "

- 3. From the title I thought it would be a rather different paper. I do not think you have answered the question 'how can we understand. . .', I think you have simply performed an analysis of the surface response. I would therefore change the title.

==> As mentioned above, the title has been modified as follows. "How can we understand the global distribution of the solar cycle signal on the Earth's surface?"

**Minor comments**

- P1L27: This sentence does not fit so well. In longer term studies (of centuries) solar influence on climate has been known about for a while. Do you mean just short term?

==> According to the reviewer's comment, we added the phrase "especially that of the 11-year solar cycle," for more precision.

- P2L11: 'global mean temperature' to you mean 'surface temperature'?

==> We changed to global mean "surface" temperature.

- Section 2.2: Here I would address my General Point 2.

==> We followed referee's suggestion. See answer for General Point 2.

- P4L14: A number of studies use additional regressors. Maybe a point or two on why these are OK.

==> For the MLR, we used indices which describe climate variability factors that have been demonstrated to have a significant impact in the middle atmosphere and at the surface, i.e. solar forcing, volcanic aerosols, ENSO, QBO (x2), and anthropogenic forcing, and which have been extensively used in many model and reanalysis solar-related studies [e.g., Chiodo et al., 2014 ; Mitchell et al., 2015a,b]. Some studies used additional regressors allowing to account for NAO variability [e.g. Haigh et al., 2005] or a third QBO term [e.g. Kuchar et al., 2015]. After testing, Kuchar et al. [2015] confirmed that the "The solar regression coefficient seems to be highly robust since neither the amplitude nor the statistical significance field was changed significantly when NAO or QBO3 or both of them were removed" in the stratosphere. However we notice that using the NAO index in the MLR to examine the surface climate is somewhat misleading from a physical point of view since it is quite well established that the solar signal modulates the NAO [e.g. Kodera, 2003] at quasi-decadal timescales. For instance, we repeated the MLR analysis for the surface temperature with (left panel) and without (right panel) the NAO index as a regressor. The solar regression coefficient is shown in Figure below.

[Figure]

It is obvious that the solar signal in the North Atlantic sector (quadrupolar temperature pattern around the North Atlantic basin) becomes weaker when the NAO index is added in the MLR because part of the signal is projected onto the NAO regression coefficient.

According to the comment, we added the following sentences. "The Arctic Oscillation (AO) or the North Atlantic Oscillation (NAO) is climate mode which is partly driven by

solar variability as will be shown later. Hence, it is not relevant to include its index in a MLR model which aims at examining the solar cycle effect on surface climate."

- P5L4-5: Maybe cite some papers that look at solar influence on climate using the Tiao method.

==> We added the following sentence: "The application of the Tiao et al. method can be found in several papers examining the solar signal (e.g., Austin et al. , 2008 ; Mitchell et al. , 2015b)"

- P5L27-30: It is not clear to me exactly where you refer to here. Is it literature, or is it panels a and b? If the latter, I still do not see all the features that are mentioned.

 P5L27-30: "Common features in the spatial structure of the solar signal in surface temperatures include
 i) (sub-polar regions): warming around 45° −60° N over the Eurasian continent and cooling west of Greenland;
 ii) (mid-latitudes): warming over the ocean basins around 30° −45° latitudes in the Northern Hemisphere
 (NH) as well as in the Southern Hemisphere (SH); iii) (tropics): 30 warming over the Indian Ocean and the
 central Pacific, and cooling in the East Pacific and the Atlantic, particularly in the SH."

==> The feature described in the text can be found in the Figures shown below.

[Figure]

- P6L5 2-3 years should probably be 2-4 years. In the literature it is often written in both ways, but I think if you look at the figures in the relevant literature, the signals at 2 years are as large as those at 4 years (with the signal max at 3 years).

==> Changed to 2-4 years.

- P6L11-22: Is there perhaps cross correlation in the regressors between solar and volcanic? For instance the response seems anticorrelated say at 45N which is a max in solar, and a min in volcanoes.

==> We tested the sensitivity of the solar signal to the volcanic eruptions by removing the volcanic years from the MLR (1982, 1983 for El Chichon and 1991, 1992 for Pinatubo).

[Figure]

Solar coefficients are shown above (as for Fig. 2) with (solid line, like in the paper) and without (dashed line) volcanic eruptions. We conclude that the solar signal derived from the MLR, for this variable, is not strongly affected by volcanic eruptions. The statistical significance (not shown) is also only marginally affected. Moreover Lean and Rind (2008), who used 1889-2006 period historical datasets, obtained results which are very consistent with ours (see their Fig 3).

- P6L32: A forcing of the vortex nearly always leads to a response in the NAM, so why is it remarkable?

==> This phrase simply indicates that the AO and solar signal exhibit very similar structure. According to the comment, the phrase has been modified as "It should be noted....".

- P8L15-16: The temperature response seems very large over Eurasia. Is this real? I find it hard to believe that the temperature response is over 2K. I think this needs to be investigated and discussed more.

==> Interannual variation of winter surface temperature is especially large over Siberia. So, variations of 2K are not surprising. Similar results using different datasets can be found in Chen et al. (2015).

Chen, H., H. Ma, X. Li, and S. Sun (2015), Solar influences on spatial patterns of Eurasian winter temperature and atmospheric general circulation anomalies, J. Geophys. Res. Atmos., 120, doi:10.1002/2015JD023415.

- P10L15-22: Is there a QBO in the model? How does the momentum forcing interact with the QBO. Surely at some points they will not be consistent with the H-T relationship?

==> There is no QBO in the model. It should introduce some additional variability. We think, however, that the average feature of the difference between strong and weak vortex experiments remain similar.

- P11L8-19: The authors are very sure about the casual links here. I think they need to be more speculative about the comments, or back it up with modelling evidence from their model.

    P11L8-19: " These characteristics of the surface response to stratospheric westerly zonal wind changes fit remarkably well to the global solar surface signals from observations (Figs. 1 and 9)."

==> According to the comment, the word "remarkably" has been removed. This sentence simply describes that the global feature of the tropospheric response is consistent.

- P12L13-19: So are the authors suggesting they do not believe the Haigh mechanism? I think it is still important, but the paragraph does not read that way. I would also cite Simpson et al, 2009.

==> The problem of Haigh et al. (2005) and Simpson et al. (2009) is that there is no reason provided for the tropical warming. It is difficult to attribute such warming in the lower stratosphere to direct solar forcing. To clarify our point of view, the sentences have been modified as follows.

"Lower stratospheric tropical heating was proposed as possible origin of the solar influence on the troposphere (Haigh et al., 2005; Simpson et al., 2009). However, the reason for the warming in the tropical lower stratosphere during high solar activity is unclear. It has been shown that such tropical lower stratospheric warming is associated with a downward penetration of westerly anomalies from the upper stratosphere and hence of dynamical origin (Kodera and Kuroda, 2002). "

- P13L13-16: Hood et al only show a subset of models, and not even all the coupled chemistry models from CMIP-5. Are there better (or additional) references that could expand on this point?

==> Hood et al. made a choice according to the reproducibility of the ozone variation in the upper stratosphere, which is the fundamental response to the solar spectra variation. It is natural to exclude such "unrealistic models" which cannot reproduce the fundamental solar influence. There are no other references that could expand on this point.

**Figures**

- Figure 1: I would make a and b more comparable. Use the same contour intervals and only plot of the oceans. Also use the same latitude ranges.

==> It is to show that in spite of different datasets (historical data or modern reanalysis data, sea surface or surface temperature) and different methods of analysis (composite mean or linear regression), similar results can be obtained. Note also that using the same contour intervals between panels a and b might bring some confusion because it is not exactly the same variables which are plotted (SST vs ST).

- Figure 4: There is a lot of detail in panel c, and it can't really be seen. Can you enlarge it to the size of the other panels.

==> We reduced the number of contours and also converted the vertical velocity from hPa/s to m/day to make the variation in the upper troposphere clearer.

---

## Author Comment (AC2) · 3 Jun 2016

Many thanks for your comments and suggestions on our manuscript. Here is our answers ( ==> ).

Anonymous Referee #2

This paper studies the 11-year solar cycle signal in Earth's surface using historical datasets and the surface evidence is further supported by the zonal mean vertical profile using ERA-Interim and previously archived model simulations. The authors present many surface and zonal mean quantities that are composited between high and low solar years. Although majority of the results presented here are known or previously published, this paper could still be informative because it provides an up-to-date and comprehensive summary of the atmospheric response to the 11-year solar cycle in the observational data sets.

The authors have attempted to examine the dynamical mechanism by which the 11 year solar cycle signal is transmitted from the tropical upper stratosphere to the surface. They suggest that the observed surface signals are largely resulted from circulation changes in the upper stratosphere through downward migration of zonal mean anomalies and changes in the stratospheric mean meridional circulation. The authors' argument on this point is demonstrated mostly by using a model simulation where westward and eastward momentum forcing was applied to the entire column of the winter stratosphere polar vortex. The initial solar UV forcing however normally confines to the subtropical upper stratosphere, i.e. above 10 hPa. Thus, it differs significantly from the strong and weak polar vortex cases in their model simulation. Firstly, the solar UV effect at lower latitudes must be transmitted to middle to high latitude to produce a definite stronger vortex, which is not always easy in the real atmosphere. This is clearly demonstrated by the different responses in the SH and NH during winter sessions.

The authors present no diagnostics of the wave-mean flow interaction or meridional circulation in the stratosphere based on observation or reanalysis data. Only if the wave forcing diagnostics from reanalysis data sets match those from their model simulations, the proposed mechanism can then be stated as the main mechanism for the solar signal seen in the SSTs or SATs. I therefore find that this part of the paper is not entirely convincing. The rather strong statement made by the authors about the cause and effect regarding the link between the surface signal and this mechanism should be tuned down and presents as one of the contributing mechanisms instead. If not, please provide additional supporting wave-mean flow interaction diagnostics using ERA-Interim or other reanalysis data sets.

The results are appropriate for ACP and the structure of the paper is sound. The clarity of paper may be improved by reducing the lengthy discussion. I have several specific comments that I would like to see addressed before the paper is published.

==> The aim of the present study is to understand the global distribution of the solar signal on the Earth's surface. Therefore, stratospheric processes, such as wave-mean flow interaction are not investigated in the present paper, but results of previous studies are refereed.

**Major comments:**

- Lines 22-24. Abstract. As I stated previously, these statements are too strong given the momentum forcing applied in the model simulation differ largely from the actual solar UV forcing.

==> The sentences have been modified according to the reviewer's comment:.

"Although the momentum forcing differs from that of solar radiative forcing, the model results suggest that stratospheric changes can influence the troposphere not only in the extra-tropics but also in the tropics through i) a downward migration of wave–zonal mean flow interactions and ii) changes in the stratospheric mean meridional circulation."

- 2. It appears to me that the atmospheric or tropospheric response in their model simulation (Figure 10) can only explain the early winter behaviour of the solar signal in the NH. It fails to explain the high latitude warming signature in the late NH winter and in SH spring and no signal in SH winter (Figure 6).

==> When a stronger westerly jet extends from the stratosphere to the troposphere in late winter, tropospheric planetary waves propagate upward along a stronger westerly jet. Then, zonal winds in the upper stratosphere are decelerated and a warming occurs in the polar middle stratosphere. This means that polar warming in late winter is rather a stratospheric response to a tropospheric circulation change. Here, we focus on the downward penetration of stratospheric influences. Therefore the absence of this feedback from the troposphere is not crucial to understand stratospheric impact on the troposphere.

- 3. Lines 5-30, Page 4. MLR may be quite useful in studying a system in which the dependent variables are linearly related to the predictors in time. The assumption may hold for annual mean fields but will not be applicable for the seasons where nonlinearity dominates. In NH winter, for instance, the authors have suggested that the stratospheric response to the 11-year solar UV cycle in early and late winter flips the sign. This suggests nonlinearity and may result in cancellation of solar signal there when a linear regression model is applied. It would be helpful to the readers if the authors make this point clearer.

==> We agree with the reviewer that the use of the MLR to derive seasonal signals is not always relevant due to nonlinear processes and requires additional care. Indeed, it is shown that the stratospheric 11-year solar cycle response rapidly evolves in the Northern Hemisphere winter (Fig. 5). In this case, where the seasonal march is crucial to understand the physical processes leading to the propagation of the solar signal throughout winter, we show only individual months and not the seasonal signals. In our study, seasonal signals are essentially shown for atmospheric and ocean surface variables to focus on the seasonal variation of climate variability modes. The only exception we made for the stratosphere is in Fig 6a where we show the averaged response for two consecutive months (Nov/Dec for NH and Jul/Aug for SH) which correspond to the "radiatively controlled" stage of the seasonal march of the solar signal and the signature for the two months is similar. We thus made the point clearer in the text (in section 2.2, last paragraph) Section 2.2 was also expanded to discuss the MLR limitations as requested by reviewer #1.

Finally, we also compared different MLR techniques to derive seasonal signal, i.e. by averaging the monthly fields before applying MLR and deriving directly "seasonal coefficient" (as we formerly did) vs. by first deriving the monthly coefficient and then averaging them to obtain the seasonal response. Both methods gave very similar results (see below).

[Figure]

Figure. DJF-averaged solar regression coefficient of the surface temperature for two different seasonal MLR methods: (top) averaging the monthly fields before applying the MLR and (bottom) deriving the monthly coefficients first and averaging them to obtain the seasonal signal.

- 4. From Figures 5 and 6, it is not clear to me how the surface temperature and circulation patterns are so-surely linked to the stratospheric circulation anomalies, as the way presented by the authors. In both hemispheres, little solar signals can be found in the polar temperature during middle winter (see Figure 6). In the NH, the mid-latitude troposphere and lower stratosphere show to be weakly warm in Nov-Dec, Jan and Feb while the polar region flips from cold to warming from Nov to Feb. Thus, why the upper level "causing" signals are effectively weaker than the "responding" signals near the surface? Or to what extent these winter temperature anomalies shown in Figure 6 contribute to the annual mean anomalies shown in Figures 1 and 4?

==> To understand the solar signal, the overall features are investigated by combining the tropospheric part of Figs. 4, 5 and 6 in Fig. 14. The solar signal in the tropospheric temperature field is relatively small in January in the NH and September in the SH (Fig. 6; Fig. 14b). It should be noted that this is a period of transition; the tropospheric temperature signal is induced by the downward penetration of zonal wind anomalies. Therefore, more statistically significance should be expected in the zonal wind field (Fig. 5; Fig. 14a). A pair of warming and cooling is formed at both sides of the axjs of the zonal mean zonal wind anomaly consistent with the thermal wind relationship. Therefore, the temperature signal is physically consistent even though the statistical significance is low. It is also shown in Fig. 7 that the surface temperature signal induced during the winter can be maintained and even amplified through an interaction with the ocean. Therefore a stronger statistically significant signal is found in the annual mean temperature field. In contrast, the annual mean zonal wind signal is less significant (Fig. 4). A possible role of ocean feedback to enhance stratospheric impact is also discussed in Yukimoto and Kodera (2007) and Misios and Schmidt (2013). The active role of the ocean is also found in the model experiment in Fig. 10 that although no external forcing

is applied in the summer hemisphere, anomalous mid-latitude warming and wave activity persist in the troposphere, in particular in the SH.

[Figure]

The above text and Figure were added in the revised version.

- 5. Line 31, section 3.4, page 8. Tropical solar signals appear to be important in this paper and the authors have devoted an entire subsection for it. However, in the abstract, it states "no warming in the tropics". Somehow, I feel that the authors need to provide the reason as why the tropic solar signals need to be specifically discussed given the most significant solar signals are found in the middle latitudes (See figure 1). Also, in what way the tropical solar signals are connected to the dynamical mechanism by which the 11 year solar cycle signal is transmitted from the tropical upper stratosphere to the surface?

=> It is rephrased as "no overall tropical warming". The amplitude of the temperature variation is small in the tropics. However, in the tropics, change in precipitation (or vertical velocity) is much more important. Figure 4c indicates a shift of the raising branch of the Hadley circulation, of which importance is evident.

A possible process producing a tropical tropospheric effect is described in the text Page l0 line 24-30 of original paper: "Previous model studies (Thuburn and Craig, 2000; Kodera et al., 2011) showed that changes in stratospheric meridional circulation affect tropical convective activity through changes in static stability in the tropical tropopause region (Eguchi et al., 2015). In the present experiments also, suppression of equatorial ascending motion occurs in the troposphere in connection with the reduction of stratospheric mean meridional circulation change, as can be seen in the residual circulation differences in Fig. 10c.

- 6. Figure 12c is rather sudden and thus potentially confusing because the wave forcing and residual circulation anomalies in late winter are not supported by any of the analysis presented earlier in the manuscript based on either data or model simulations.

=> This is based on the results in Kodera and Kuroda (2002) and Matthes et al. (2006).

The sentence has been modified as follows."we show these two stages schematically in Fig. 13 based on previous studies (Kodera and Kuroda, 2002; Matthes et al., 2006; Matthes et al., 2013) ".

- 7. Lines 5-8, page 13. I cannot see the reason why a longer lasting radiatively controlled stage in the subtropical SH upper stratosphere can lead to an anomalous weakening of the stratospheric jet and warmer polar stratosphere (Figures 5 and 6). It appears to me that the argument based on dynamical versus radiative control is definitely valid in part but it remains not sufficient to explain all the stratospheric anomalies.

==>In the SH, a weakening of the stratospheric jet and a warmer polar stratosphere becomes evident in September "near the equinox", when differential solar forcing becomes small. Then, planetary waves propagate in weaker winds in the stratosphere and produce polar warming in October.

- 8. Lines 21-34, page 14. These sound much like results rather than discussion and concluding remarks. Suggest moving to an earlier section instead. As I have stated before, the composite difference estimated from the simulated weak and strong polar vortex are not exactly representative to actual solar UV forcing. First, the solar UV forcing has much smaller magnitude. Second, the solar UV effect is located much higher in altitude than the model simulation assumed. As a result, the solar UV effect should be much weaker than what has been suggested by the model simulation.

==> According to the reviewer's comment, this part has been moved to a new section 5. Centennial scale variation. To conform to this change, the following sentences are added in Introduction and Discussion.

Introduction
"To get insight into a centennial solar variation such as the Maunder minimum, the effect of centennial scale stratospheric circulation changes on the troposphere is briefly studied in section 5."

Discussion
"It should also be noted that centennial circulation changes produced in the stratosphere can affect global mean surface temperature through changes in the Earth's surface condition without changes in total solar irradiance."

- 9. Some of the fields are quite messy (e.g. Figure 4b,c; Figure 6) or not statistical significance is shown (e.g. Figure 1a). Some of the features are not statistically significant but are discussed as the cause for the surface anomalies. I suggest that the discussion around these figures/features needs to be more careful.

==> As discussed in the paper, solar signal is characterized by its global distribution. We consider that we should not put too much importance on local variables. In this respect, the way that Zhou and Tung (2010) made to test the statistical significance of the global solar signal as in Fig. 1a, may be better adopted to this problem.

Minor comments:

- Line 11, abstract. "no warming in the tropics". This is not clear. "No warming" could imply either "cooling", "no signal" or "complex signal with longitudinal variation".

==> According to the comment, the phrase was modified as "no overall tropical warming".

- 2. Line 14, abstract. "the subtropical jet". The term is not clear. The subtropical jet in the atmosphere often refers to the tropospheric subtropical jet. Here, the authors refer to the upper stratosphere subtropical jet. Climatologically speaking, there is no subtropical jet in the stratosphere anyway. There is only one jet in the stratosphere which is the polar vortex which initializes at lower latitudes in early winter.

==> Study on the subtropical jet in the middle atmosphere is rare and may not be well known. We therefore introduced the following explanation and figure about two different nature of westerly jets in the middle atmosphere.

" It should be noted that there are two kinds of westerly jets in the middle atmosphere. Figure 13 displays the climatological poleward temperature gradient during winter solstice (Jun in the SH and December in the NH). The meridional temperature gradient is large in the subtropics of the upper stratosphere due to solar UV heating, while in the lower stratosphere, the gradient is large in the polar region due to strong longwave cooling. They are respectively connected to the subtropical and polar night jet. Poleward and downward penetration of solar signals in the middle atmosphere occurs through interaction between these jets and planetary waves propagating from the troposphere."

[Figure]

- 3. Line 1, page 2. "amplify" -> "act to amplify".

`several mechanisms have been proposed that amplify the initially small solar effect`
==> Corrected as "act to amplify"

- 4. Line 13, page 2. Zhou and Tung (2010) not cited in the reference list.

==> It is located at the end of the reference list.

- 5. Line 26-27, page 2. "Because solar signals in SLP data are inconsistent, probably due to the temporal and spatial limitations of the data, we instead study pressure or geopotential

height fields . . .". It is confusing firstly because the SLP is pressure, isn't it? Also, it is known that solar signal tends to wax and wane with the different periods under consideration. Would it be better that we admit that we still do not understand why it happens rather than blaming the data quality. The wax and wane can also be found in modern data sets such as ERA-40 or ERA-Interim.

==> According the comment, we rewrote the sentences as follows.
" Because sea surface temperature (SST) is more persistent than the sea-level pressure (SLP), long-term variations can be more easily detected in the temperature field. Therefore, we investigate mainly surface temperature variation from the historical data, complimented by pressure or geopotential height fields with a modern dataset. "

- 6. Line 16, page 4. "predictorand" -> "predictors".

==> corrected

- 7. Line 19-20, page 7. "The differences in the latitudinal structure of the warming suggested. . .".   This is not clear especially from the annual mean field. These statement can only be said when other dynamically quantities are also analysed. Suggest to remove or cite references to support such claim.

==> According to the comment, the sentences have been modified as follows.
"Previous studies suggest that the solar signal in the tropical lower stratospheric temperature is mainly induced through a modulation of the stratospheric mean meridional circulation or the Brewer-Dobson circulation (e.g. Kodera and Kuroda, 2002; Hood and Soukharev, 2012). Inspection of Figs. 5 and 6 reveals that the warming in the middle and lower stratosphere is produced in association with very sharp zonal wind anomalies. In fact, such strong meridional gradients of the zonal winds could not be produced by latitudinal difference of the radiative heating rate which mainly depends on the solar zenith angle."

---

## Author Response (AR2)

**Response to Editor**

Many thanks for taking care of our paper and the comments. Here is our response.

**Co-Editor Decision: Reconsider after major revisions** (27 Jul 2016) by Peter Haynes Comments to the Author:**

Both referees have provided comments on the revised version of your paper. One referee now recommends publish after minor revision. The second referee remains more critical. I have looked quite carefully at the revised paper myself and I have some sympathy with the general views of the second referee -- 'When come to the discussion, the authors need to take good care about what has been done previously and what kind of new insight is provided by their analyses here.'. (The other referee expresses similar views, but is more tolerant.)

Therefore please can you provide a revised version of the paper which takes account of the detailed comments of the referees, and also of my own (Editor) comments below, along with a set of responses that make it clear how the various comments have been addressed (or why they have not been addressed). Clarity on how exactly the paper relates to previous work and careful distinction between, for example, clearly described physical mechanisms and apparent connections between different atmospheric phenomena will help ensure the paper is positively received. I hope to be able to accept the next version of the paper for publication in ACP without further consultation with referees, but that depends on how thoroughly these various concerns have been addressed.

According to the comments, a major revision of the text has been made to improve the clarity of the paper:

- i) In the discussion section, we added a description of the results of previous studies on surface solar signal using historical temperature data and made a comparison with the present study.
- ii) The discussion section (section 6) has been reorganized into two parts: 6.1 Tropospheric processes and 6.2 Connection with the stratosphere.
- iii) We added a schematic figure (Fig. 16) to characterize tropospheric processes related to the solar signal.

We hope that these changes will make our paper more understandable.

**EDITOR COMMENTS:**

p1 I16: 'whereas in the Southern Hemisphere the subtropical jet plays the major role' — my reading of the text describing Figs 5 and 6 is that this is the 'UPPER STRATOSPHERIC' subtropical jet — this should be stated explicitly here.

Modified as suggested.

p2 128: 'Surface temperature and pressure have been measured for more than 100 years. Thus, the relationship between surface temperature variations and solar activity can be investigated using a global historical dataset.' — the logic here works only if solar activity has also been measured for the required time — it might be data on solar activity that limits the period over which the relationship can be investigated. In the following text you mention datasets such as 'proxy solar irradiance' from Lean et al (1995) but give no information at all (not even a few words) on the limitations of such datasets.

Direct measurement of the solar UV is only recent, but the sunspot numbers, which is a proxy of the solar EUV is available from the 18 century. The solar extreme ultraviolet (EUV) produce the ionization in the Earth's upper atmosphere. Therefore change in the solar EUV radiation is felt on the Earth's surface as change in geomagnetic field induced by the electric current in the ionosphere. It is, thus, possible to associate the variation of sunspot numbers to the solar EUV activity. Comparison of the variation calculated from Earth's magnetic field demonstrates excellent agreement between the 10.7 cm solar radio flux (F10.7) (Svalgaard, 2007). Therefore we can use sunspot numbers as a proxy of the solar EUV variation.

Roughly speaking, total solar irradiance (TSI) measured from the space, shows in phase relationship with the 11-year cycle of the sunspot numbers, although centennial scale variation in the TSI is still unknown. The proxy solar irradiance data of Lean et al. (1995) in question, is used in the study of Lohmann et al. (2004), but not in the present study. It should be noted that we refer only to the part of their result related to the 11-year solar cycle obtained by removing the long term variation by applying 9–5 year band-pass filter. We, therefore, consider that there is no need to give further detailed information on the proxy solar irradiance data used in Lohmann et al. (2004).

The explanation about the sunspot numbers and F10.7 are included in revised text.

p6 l24: 'the results show a similar pattern' — there are similarities but there are also differences. In the paragraph below you enumerate the similarities — why don't you enumerate the differences?

The interference between the solar signal and regional teleconnection pattern can produce temporal variation of the relationship. Therefore, it is necessary to ensure that the solar cycle relationship found in a short dataset should also be presents in long historical data set.

It is of course important to address the difference, but it is a very difficult problem. Before this, we first need to understand stable response. According to the comment, we added the following sentences in the text. " In spite of overall similarities, large differences can be found in some regions, such as over the subtropical eastern Pacific east of Hawaii. It shows large warming in the historical data (Fig. 1a), but cooling in the modern era data (Fig. 1b). It is however difficult to identify whether the difference in short-term data is merely due to statistical fluctuations, or related to a change in basic climatological states caused by other factors, such as global warming, or ocean circulation change, etc. Here, we concentrate on the stable solar response to first understand how it is produced at the Earth's surface."

Fig 2: You show surface temperature signals associated with different external drivers, but have made a particular decision re choice of units. What does '(for 100km)' mean in (b). What are the units for (f)? Wouldn't it make more sense to have (b) + (c) either at the beginning or end of the sequence (since those quantities are nothing to do with regression)?

For the CO2, the trend is scaled on 54 ppmv increased of CO2, which corresponds to the increase between 1979 and 2010 (from 336 to 390 ppmv). The scale of CO2 variation is indicated in Fig. 2.

(b) is meridional gradient of the SST: temperature difference for 100 km of distance. Figures have been reorganized according to the suggestion. Climatological SST, and its meridional gradient are placed at the end.

p5 l34: Shouldn't 'levels' be 'measures'?

It was rephrased as "high and low levels" of solar activity.

p7 110: 'To identify the physical mechanisms responsible for the solar surface signals, a comparison of the surface temperature pattern associated with other forcings has been performed.' — of course this 'identification' can only be indirect. More seriously you have previously said that the AO is a 'climate mode'. But here you are classing it as a 'forcing'.

I think there is some misunderstanding. We do not include the AO as forcing, like ENSO, Volcanic aerosol, or CO2. The AO is discussed in the following paragraph as one "mediates" solar signal from the stratosphere to the surface.

p8 I20: 'Previous studies suggest that the solar signal in the tropical lower stratospheric temperature is mainly induced through a modulation of the stratospheric mean meridional circulation or the Brewer-Dobson circulation' — followed later by 'in the middle to lower stratosphere has a dynamical origin'. You seem to be saying the same thing twice — but I suppose that you are confirming that the suggestion of previous studies is consistent with your analysis presented here. Please clarify to confirm logic.

According to the comments the sentences have been modified as follows. "The narrow latitudinal extent of the zonal mean zonal wind anomalies at mid-latitudes of the middle-lower stratosphere in Fig. 5 is difficult to explain only from a radiative heating change. The differences in the latitudinal structure of the warming suggest that the warming in the stratopause–upper stratosphere has a radiative origin, while for the second warming in the middle to lower tropical stratosphere, dynamical process plays an important role as suggested in previous studies (e.g. Kodera and Kuroda, 2002; Hood and Soukharev, 2012)."

p11 I15: 'See Yukimoto and Kodera (2007) for a more detailed description of the experiments.' From my point of view there are some points that are obscure in YK2007 which need saying explicitly here. For one thing, the formula you present on p944 of YK2007 seems inconsistent with what you say in the text. The formula includes a cosine function of time, the text says 'applied only in the winter hemisphere'. That seems inconsistent. Also is the force applied in both NH and SH winter hemispheres, or only in one? That also wasn't clear to me from YK2007. A important point that you mention in YK2007 but don't mention explicitly here — although it is more important here, is that the model results are for a forcing that is applied (in winter) over a period of 50-100 years. So there is a real question about whether the response will be similar to the solar cycle response (where the relevant period would be about 5 years) because, for example, the ocean response may significantly different. My view is that this should be made explicit in the paper under consideration. In the caption to Fig 12 you note that the experiment — if we ignore the question of how the variation in incoming solar radiation gets converted into something equivalent to an applied force — is comparable to an 'extended period of extreme solar minimum (Maunder Minimum-like)'. You should say this in the text too — and perhaps in the abstract — bearing in mind that the observational results in the paper are based on the 11-year cycle.

In fact, the forms given in YK2007 are only for the NH. We gave the forms valid for both hemispheres as follows.

 $F_m = A_0 f(p)(\sin 2\varphi)^2 \text{ MAX}\{0, \cos [2\pi(n-n_0)/365]\}$

with  $A_0$ : maximum amplitude (5 m s-1/day), *n*: day of the year, and  $n_0$ : central day of the winter

(15 Jan in the NH, and 15 Jul in the SH).

The vertical profile f(p) is expressed as,

$$\begin{array}{ll} f(p) &= 1 & p < 10 \\ &= \ln (p/100) / \ln (0.1) & 10 < p < 100 \\ &= 0 & p > 100 \end{array}$$

where *p* denote the pressure (hPa) and  $\varphi$  the latitude (radian).

According to the comment, the following sentences have been added in Abstract and text. -Abstract:

Model integration of 100 years of strong or weak stratospheric westerly jet condition in winter may exaggerate long-term ocean feedback. However, the role of ocean in the solar influence on the Earth's surface can be better seen.

-Text

p. 13

Model integration of 100 years of strong or weak stratospheric westerly jet condition in winter may exaggerate long-term ocean feedback. However, the role of ocean in the solar influence on the Earth's surface can be better seen.

-p. 14

Note that the results of this experiment are more comparable with an extended period of extreme solar minimum (Maunder Minimum-like) conditions.

p15 I10: 'tropospheric solar signals in the SH do not resemble the SAM, which is related to variability in the polar-night jet' — whilst it might be true that the SAM can in part be associated with variability in the polar-night jet this is surely not the case for ALL SAM variability? So the logic here — apparently 'there cannot be an SAM signal because there is no polar-night jet signal' seems flawed.

The phrase has been modified as follows. "tropospheric solar signals in the SH do not resemble the SAM, which is sensitive to the variability in the westerly jet in high latitude."

**Answer to reviewer 1**

Thank for your reading of our manuscript and the comments. Here is our responses.

Reviewer 1

The authors have done a good job at responding to my comments. I note two comments that should still be addressed to a higher standard, but they are only minor.

1. Title: I still do not agree the current title makes sense. Asking 'How' implies that your conclusion will be about 'how we can understand', but your conclusions are not about 'how' they are about 'what'. You main conclusions are not anything like 'We can understand solar signals on the surface using this novel new technique, and this is the point of the paper', your conclusions are 'we have some novel results, any the methodology of the paper is not the main point'.

If we understand correctly, reviewer #1 does not agree with the fact that we are using "How" because it would imply that we provide new results from a new methodology and not only new results from a new interpretations based on existing and extensively used methodology.

In this paper we discuss the processes on how solar signals can be produced in different regions over the globe. Therefore, the title, "How can we understand the global distribution of the solar cycle signal on the Earth's surface?" seems appropriate.

2. I do not agree with the newly added sentence '... their limited interpretive capacities make it difficult to study physical mechanisms.' I think this sentence would be better said as '... their advanced statistics is at the expense of allowing analysis as smaller scales.'

Unfortunately we cannot understand the meaning of the sentence in the comment: at the expense of allowing analysis as smaller scales." Could you, please, explain why this is better?

**Answer to reviewer 2**

Thank for your reading of our manuscript and the comments. Here is our responses.

**Reviewer 2**

The authors have partly incorporate my previous comments. However, the revised manuscript remains not entirely convincing and sometimes even more confusing. In particular, they still insist that "diverse aspects of the solar signal on the Earth's surface can be explained solely by solar UV heating changes in the upper stratosphere which penetrate to the troposphere …" (lines 28-29, Page 16). However, their own analysis and other papers suggest that the ocean-air interaction and a change in ITCZ positions do contribute or even amplify the solar signal. This type of writing could cause a great amount of confusing in the literature. In addition, the discussion around the mechanisms remain very speculative but the speculative view is presented with even greater certainty. I also notice that the authors chose to use different periods of ERSST data to produce Figures 1, 7 and 8. This needs to be explained as why different periods were used for different diagnostics and sensitivity of the results in association with the periods chosen.

The clarity of paper must be improved by reducing potential confusion in terms of the responsible mechanisms. When come to the discussion, the authors need to take good care about what has been done previously and what kind of new insight is provided by their analyses here. Also, they need to carefully differentiate the cause and the response of the solar signals. I have several specific comments that I would like to see addressed before the paper is published.

The solar signals in the surface temperature found by different authors in previous studies have been added in the text. The aim of the present study is to find out how these solar signals in different regions on the globe are interrelated and how they are produced. Because of strong feedback in the climate system, generally it is difficult to separate the cause and response. Therefore, the results of idealized experiment are invoked to understand the processes producing solar responses in the troposphere.

According to the comments, a major revision of the text has been made to improve the clarity of the paper:

- i) In the discussion section, we added a description of the results of previous studies on surface solar signal using historical temperature data and made a comparison with the present study.
- ii) The discussion section (section 6) has been reorganized into two parts: 6.1. Tropospheric processes and 6.2. Connection with the stratosphere.
- iii) We added a schematic figure (Fig. 15) to characterize tropospheric processes related to the solar signal.

We hope that these changes will make our paper more understandable.

**Specific comments:**

1. Lines 16, page 2. I think that CMD projection method is not as simple as "taking the composite difference between periods of high and low solar activity during the 11-year cycle". Please clarify.

As requested we clarified the CMD method as follows:

"However, Zhou and Tung (2010) extracted a global spatial pattern of sea surface temperature (SST) variations associated with the solar cycle by applying a composite mean difference (CMD) projection method, particularly relevant to estimate the robustness of a global spatial signal. This method segregates data into groups of high and low solar activity during the 11-year cycle. A global spatial pattern is then obtained from the composite-mean difference between the high and low solar group. Finally, the original data is projected onto this CMD spatial pattern, resulting in a time series. The method is successful when the correlation between the resulting time series and the solar forcing is high."

2. Line 25-26, page 2. The authors need to explain exactly what new knowledge this paper can add to the existing literature. For instance, please explain why investigating "processes producing solar signals on the

Earth's surface in a specific region" cannot lead to "the overall understanding of solar signals on the entire Earth's surface".

We included in the revised text the description about the solar signals so far found by different authors. These authors, however, do not discuss a possible atmospheric process producing positive and negative solar signals in different regions. We show that diverse aspect of the solar signal can be more easily understood by grouping the response according the characteristics of climate zones: warming in oceanic frontal zone in mid-latitudes, but cooling in tropical cold tong regions. Such "global view" is necessary to understand the solar signal.

3. Lines 22, Page 2. "solar surface signal" is a strange phrase and potentially misleading. Consider improving. "solar signal near the surface"?

The phrase has been modified as "surface solar signal".

4. 2.1 Data section, page 4. The authors need to explain why three different time periods of c data were used to produce Figures 1 (1854-2007), 7 (1880-2010) and 8 (1890-2012). How sensitive the results would be if a single period was used for those diagnostics?

Gridded monthly data ERSST starts from January 1854. However, data are very sparse before1880. Zhou and Tung (2010) made two calculation using data from 1854 to 2007 and from 1880 to 2007. Resultant solar signals, reproduced below, are very similar.

Figs 2 and 3 in Zhou and Tung (2010)

In their analysis, however, the data during world war II have been omitted due to bad quality. We repeated their analysis including all data, but the result is unchanged as can be seen in our Fig. 1. Therefore, solar signal in ERSST is not sensitive for small difference of the period.

For Fig 9, the lagged solar response is highly consistent if we start from 1870, 1880 or 1890. For shorter ERA period (1979-2010) it is still consistent, even if the mid-latitude warming is amplified and the Labrador sea cooling less strong. To show the consistent features, both lagged solar responses for long (1880-2010) and short (1979-2010) periods are shown in Fig. 9.

For the EOF analysis, better quality of the data is requested. In the tropical region where the number of observation is limited, we used the data after 1890 (Fig. 10). Note that another global SST dataset COBE starts only from 1891. The following explanation has been added to the text, "For the EOF analysis, the period 1890-2012 is chosen for better observational coverage (Smith and Reynolds, 2003) ".

5. Lines 10-13, page 4. It is better to state that the period used for the analysis is 1979-2010, rather than "1979 to the present".

Corrected according to the comment.

6. Lines 7-8, page 5. "Since the ERA-I dataset only starts in 1979, the QBO regressors are removed from the MLR when the long-term historical dataset (i.e. ERSST) is analysed." This sentence does not make sense for the following reasons.

1) Do you get consistent result if you remove the QBO regressors from ERA-I analysis? Namely, when you use MLR for the 1979-2010, would the results, especially the tropical lower stratosphere warming signal, change if you remove the QBO regressors?

We have verified that the inclusion or removal of stratospheric QBO regressors little affects the solar component, in particular in the troposphere, in the MLR of 1979-2010 as can be seen below. This is indicated in the revised text. Adding to this, correlations among different regressors generally diminish when the data length become longer. Therefore, the solar component may be little affected by the exclusion or inclusion of the QBO term in longer dataset.